# Numerical and Experimental Study of Mode Coupling Due to Localised Few-Mode Fibre Bragg Gratings and a Spatial Mode Multiplexer

**DOI:** 10.3390/s25196087

**Published:** 2025-10-02

**Authors:** James Hainsworth, Adriana Morana, Lucas Lescure, Philippe Veyssiere, Sylvain Girard, Emmanuel Marin

**Affiliations:** 1IRT Saint Exupery, B612, 3 Rue Tarfaya, 31400 Toulouse, France; 2Laboratoire Hubert Curien UMR 5516, Institut d’Optique Graduate School, Université Jean Monnet Saint-Etienne, CNRS, 42023 Saint-Etienne, France; 3Ministère de l’Enseignement Supérieur et de la Recherche, Institut Universitaire de France (IUF), 1 Rue Descartes, 75005 Paris, France

**Keywords:** few-mode optical fibre, fibre Bragg grating, mode conversion, spatial mode multiplexer, modal filtering, mode-selective excitation, localised FBG inscription, mode coupling, optical fibre sensing, wavelength division multiplexing

## Abstract

Mode conversion effects in Fibre Bragg Gratings (FBGs) are widely exploited in applications such as sensing and fibre lasers. However, when FBGs are inscribed into Few-mode optical Fibres (FMFs), the mode interactions become highly complex due to the increased number of guided modes, rendering their practical use difficult. In this study, we investigate whether the addition of a spatial mode multiplexer, used to selectively excite specific fibre modes, can simplify the interpretation and utility of few-mode FBGs (FM-FBGs). We focus on point-by-point (PbP)-inscribed FBGs, localised with respect to the transverse cross-section of the fibre core, and study their interaction with a range of Hermitian Gauss input modes. We present a comprehensive numerical study supported by experimental validation, examining the mechanisms of mode coupling induced by localised FBGs and its implications, with a focus on sensing applications. Our results show that the introduction of a spatial mode multiplexer leads to slight simplification of the FBG transmission spectrum. Nevertheless, significant simplification of the reflection spectrum is achievable after modal filtering occurs as the reflected light re-traverses the spatial mode multiplexer, potentially enabling WDM monitoring of FM-FBGs. Notably, we report a novel approach to multiplexing FBGs based on their transverse location within the fibre core and the modal content initially coupled into the fibre. To the best of our knowledge, this multiplexing technique is yet to be reported.

## 1. Introduction

FBGs are passive in-fibre optical components which can be applied to a range of fields such as sensing, fibre lasers, and communications. The FBG finds its uses due to a spectral feature called the Bragg resonance, a narrowband reflection centred around the so-called Bragg wavelength. The location of this Bragg wavelength depends on both the effective refractive index of the modes of propagation within the fibre, and the period length of the grating structure. Thus, these features can be manipulated to create useful components, such as cavities for fibre lasers [1] or notch filters at specified wavelengths [2]. These two variable parameters are also sensitive to changes in the FBG environment, meaning observed shifts in the Bragg wavelength can be interpreted for sensing physical parameters including, but not limited to, temperature and strain [3]. As a result, FBGs are becoming common tools in structural health monitoring [4,5] and show promise for use in complex fields such as aerospace [6].

In most cases, single-mode fibres are the preferred choice in which to inscribe an FBG. Their small core diameter, at 8.2 μm for standard SMF-28, and the relatively low refractive index difference between the core and the cladding means that the fundamental mode is the only supported mode in the fibre core. A direct result of this is that the SM-FBG reflection and transmission spectra remain simple, with a perfect uniform SM-FBG producing just one peak and trough in the reflection and transmission spectra, respectively. Due to this single-peak reflection spectrum, several FBGs can be written into a single fibre, each with a differing grating period. This FBG monitoring method, called wavelength division multiplexing (WDM),allows for several sensors to be inscribed within the same fibre for quasi-distributed sensing applications.

Nevertheless, while a simplistic spectral output is valuable for many applications, multimode behaviour and modal energy exchange in FBGs are key traits necessary for a multitude of further applications. To enter into this field of research, the fibre must support multiple modes of propagation. While SM-FBGs support one core guided mode, above its cut-off wavelength, they still show limited, yet highly useful mode conversion applications. For example, tilting the FBG with respect to the transverse fibre cross-section forces light to reflect into the fibre cladding. If the coating of the fibre is removed, the cladding is able to support multiple discrete modes. This results in the appearance of many energy exchange peaks in the transmission spectrum as energy from the fundamental mode is coupled to the cladding modes. This has found extensive uses in mechanical and biochemical sensing [7], and plasmon resonance excitation for refractometric analysis and biochemical sensing [8].

A similar outcome to that observed in tilted FBGs can also be found in localised FBGs. The FBG Refractive Index Modification (RIM) does not span the whole transverse centre of the fibre core and when they are located away from the transverse centre of the fibre core, the circular symmetry of the core is broken. This then also allows for the core guided mode to be coupled to the fibre cladding modes [9,10,11].

However, while not commonplace, FBGs can also be inscribed in both Few-mode Fibres (FMFs) and Multi-mode Fibres (MMFs) alike. These fibres employ a mix of increased core size, reaching 62.5 μm in OM1 MMF, for example, and increased core-cladding refractive index difference, to allow for the presence of several or large numbers of core guided modes. This increased number of modes results in more resonant peaks present in the FBG spectra, spanning a large bandwidth. Due to this complex spectrum, these fibres are rarely used for sensing applications as they limit the multiplex-ability of the FBGs.

Nevertheless, the addition of FBGs to FM and MMFs has sparked interest, as the presence of multiple modes sets the stage for more complex FBG-induced mode conversion effects and use cases. This could open the door for fibre-mode filters for communication purposes [12], fibre lasers with controllable mode profiles [13], and a multitude of sensing applications including multi-parameter sensing [14].

This has led to several studies aimed at better understanding the complex modal coupling within the fibre and how to effectively control the phenomena of FBG-induced modal energy exchange. Citing several, but not all of these studies, Dostalov et al. observe how the geometry and transverse position of a localised FBG may be used to control the modal coupling effects within a multimode fibre [15]. The presented work depicts how these factors may effect the resonant dip amplitudes relating to specific mode coupling.

Ali et al. [16] investigated phase mask-inscribed FM-FBGs when modes are selectively coupled to the fibre at its input using a phase plate. In this work, they found that by selecting the input modes, it is possible to excite specific modal self- and cross-coupling effects, modifying the output spectra.

Li et al. [12] propose a mode add-drop technology based on FM-FBGs for Mode Division Multiplexing (MDM) in optical communications. This technology uses localised FBGs written with different transverse eccentricities as filters for specific modes. Once the filter is applied, the specific mode the filter is designed for is reflected, removing it from the signal, while all other modes pass over the filter relatively unperturbed. The removed mode can then be re-added at will.

In this work, we expand on these studies in an attempt to further simplify the use of FM-FBGs for sensing applications. Our study is achieved by investigating the extents of spectral control possible when localised FM-FBGs are used in tandem with a spatial mode multiplexer. Localised FBGs are already showing their importance in selective coupling to higher-order modes [9,10,11,12,15]. However, the application of a spatial mode multiplexer provides additional freedom to investigate and tailor modal cross-coupling effects. This is due to the fact that the spatial mode multiplexer allows the user to selectively control the modal composition initially excited within the fibre. We aim to investigate how combinations of initial mode excitations and localised FBG transverse position can affect the overall spectral output of the system. Due to the large number of possible FBG transverse locations and initial modal excitation conditions, we present an extensive numerical study supported through experimental validation.

## 2. Achieving FBG Spectral Control

### 2.1. Proposed Setup

As previously stated, to realise selective modal excitation in localised FM-FBGs, we employ the use of a spatial mode multiplexer. The spatial mode multiplexer, in our case a Cailabs Proteus-S (Cailabs, Rennes, France) [17], allows us to control the mode, or modes, initially excited in the fibre. The spatial mode multiplexer and FBG interrogation configuration can be seen in Figure 1. The Proteus-S uses Multi-Plane Light Conversion (MPLC) to convert single-mode fibre inputs to a specific mode [18]. The Proteus device can be acquired to output either Hermitian Gaussian (HG), Laguerre Gaussian (LG), Linearly Polarised (LP) or Orbital Angular Momentum (OAM) modes. The specific device available to us while conducting this investigation was designed to output HG modes and was capable of producing the 10 HG modes listed in Figure 1.

Transmission and reflection spectra were acquired via the interrogation setup outlined in Figure 1. The ensemble is excited by a Tunics Plus Net Test 3642 HE CL sweeping laser (Yokogawa, Tokyo, Japan) connected to a circulator via a Corning SMF-28 fibre (Corning Inc., Corning, NY, USA).For each transmission and reflection spectral pair that is obtained, the circulator is connected to one of the Proteus SMF-28 input fibres. All other input fibres for the Proteus remain disconnected from the interrogation circuit. Each of the Proteus input fibres follows a different optical path within the Proteus itself. Each internal path result in the single-mode input is reformed into a specific free-space HG mode. This free-space mode is then coupled to the fibre within the confines of the device. Therefore, for each spectral sweep, we excite the Prysmian output fibre (Prysmian Group, Milan, Italy)with one single HG mode at a time.

To protect the jacketed 10 m output fibre from damage during testing, a 5 m unjacketed extension fibre was spliced to its end face. Since many FBGs with varying eccentricities were to be tested, the output fibre would have to undergo frequent cutting, cleaving, and splicing. The extension fibre then served as an expendable length of fibre, in place to absorb this splice-induced degradation, thus preserving the integrity of the main output fibre, an integrated part of the Proteus device.

The single HG mode excitation signal then interacts with the FBG spliced to the end face of the 5 m extension fibre. The transmission signal, after interaction with the FBG, is recovered by connecting the unspliced end of the fibre, in which the FBG is inscribed, to a Yenista CT400 optical spectrum analyser (Yenista Optics, Lannion, France) with rapid connectors.

The reflected signal then repasses over the 5 m extension fibre, 10 m output fibre, and the Proteus. The Proteus, now acting in its reverse configuration, acts as a demultiplexer. Therefore, the reflected signal, now consisting of a superposition of modes after interaction with the FBG, is separated into its modal components within the device itself. These components are then converted back to single-mode signals and coupled back to their input fibre of origin. As only one input fibre is connected to the interrogation setup, all other reflected signals are disregarded. However, the reflected signal corresponding to the attached input fibre reaches the circulator. Here, the signal is redirected by the circulator to the Yenista CT400 optical spectrum analyser, and the reflection spectrum is obtained.

For a given FBG, this procedure was then repeated for each HG mode input fibre to obtain a complete picture of how each possible excitation mode interacted with the attached FBG.

Although the Cailabs Proteus was used in this study, it is important to note that other spatial mode multiplexers may be used in its place. However, to obtain similar results to those presented in this study, the device should meet several key criteria. Firstly, it is important that the device in question is bidirectional. That is, the device should be capable of performing as both a multiplexer and a demultiplexer, depending on the direction of light propagation. This is imperative to obtain the reflection spectrum. Secondly, the device should be capable of working efficiently over a relatively broad wavelength range, such as the telecom C-band (1530–1565 nm). This is important if the setup is intended for sensing purposes, as the centre wavelength of the Bragg peaks will naturally drift in response to environmental changes. Additionally, if WDM of several FBGs is intended, the bandwidth of the device should account for this. The multiplexer should allow for arbitrary excitation and interrogation of specific modes or mode groups with limited crosstalk and insertion loss.

Therefore, devices that depend on multi-mode interference effects, for example, are not suitable for use in this configuration as they generally excite a fixed superposition of modes, over a limited bandwidth, and do not allow modal reflections to be interrogated separately [19]. A possible alternative to the Proteus that fits these criteria, given suitable design constraints, would be photonic lanterns using varying types of excitation SMFs to couple to specific modes or mode groups within the FMF [20]. Alternatively, other free-space mode converters make for strong candidates, such as those based on binary phase plates and beam splitters [21].

### 2.2. Localised FBG Realisation

Localised type II FBGs were fabricated in the aforementioned Prysmian fibre via the point-by-point (PbP) method, through the fibre coating, with a Pharos femtosecond laser at a wavelength of 515 nm and pulse length of ∼180 fs. The PbP method was selected because it is the only inscription method currently known that produces the highly localised RIM necessary for this study. Other inscription methods, relying on laser interference patterns, tend to cover the majority of the core transverse cross-section. This lack of localisation of the RIM thus renders these methodologies obsolete for this study.

In order to create the localised RIM, a 40x objective with a numerical aperture (NA) of 0.75, was used to focus the femtosecond laser into the core of the optical fibre. The laser beam diameter at focus, $d_{0}=2\omega_{0}$, with $\omega_{0}$ the laser beam waist at focus, can be estimated via the diffraction limit $d_{0}=0.61\lambda /\mathrm{NA}$ [22]. Similarly, the depth of focus, $L_{0}$, can be estimated as $L_{0}=2z_{r}=2\pi \omega_{0}^{2}/\lambda$ [23]. Here, $z_{R}$ is the well-known Rayleigh range. For our laser configuration, we then obtain $d_{0}=0.84$ μm and $L_{0}=2.1$ μm. Both of these values are in agreement with the phase contrast imagery of the femtosecond-induced RIM in Figure 2.

As is commonplace in a PbP inscription setup [24,25], the fibre is aligned to the laser focal plane using a high-precision stage. In our case, two Newport XMS100-S precision stages (Newport Corporation, Irvine, CA, USA) were used to control the x and y alignment, controlled by both a Newport XPS-D in conjunction with direct machine controlling software. This enabled precise alignment of the FBG within the core of the fibre along the entire inscription length of 5 mm. During inscription of the FBG, the laser repetition rate was kept at a fixed value of 189 Hz and the stage displaced in the direction of the fibre long axis at a constant velocity. The FBG period is thus dictated by the stage velocity and laser repetition rate. Each FBG was inscribed with a period of Λ∼1.6 μm, resulting in a third-order resonance peak for the fundamental mode, ${\mathrm{LP}}_{01}$, around 1555 nm.

To maintain the localised nature of the FBG RIM, the laser pulse energy was limited to a maximum value of 7.94μJ. This helped to avoid laser self-focussing effects evoked by high pulse energies (Figure 2c). Additionally, all FBGs in this study were inscribed via a single passage of the laser to prevent the formation of multiple RIMs. We observed that multiple RIMs often formed when using multiple laser passes, a phenomenon we believe to be caused by shifts in the focal point between consecutive re-passages, potentially due to the laser-induced densification of the silica (Figure 2d). This inscription setup thus allowed for accurate positioning of the FBG within the fibre core along all three axes of the fibre. Furthermore, the localised FBGs could be displaced by values of dx and dy from the transverse centre of the fibre core, following the coordinate geometry shown in Figure 2, giving each localised FBG a specific eccentricity.

### 2.3. System Simulation

As previously remarked, the proposed setup allows for a large number of FBG locations and initial modal excitation conditions. To understand the extent of spectral control possible, in this study, we simulate at length the inherent physics of each aspect of the setup outlined in Figure 1. Due to the complex nature of simulating both localised FBGs and the presence of free space to LP mode coupling upon exciting the fibre, we opted to write an overarching Matlab code which would give us the freedom to tailor the simulation to our setup—a liberty unachievable when using commercial simulation software for FBGs and optical fibres.

The numerical work carried out in this study is laid out in Figure 3. In summary, we begin by calculating the effective refractive indices of the guided fibre modes. We use this information to produce their electric field distributions. The electric field distributions for the Proteus-S are also modelled and their beam waist estimated. Using this information, we are able to estimate the power coupled to each fibre mode from a specific HG mode excitation. This also allows us to calculate the power coupled back into the Proteus-S by the FBG reflection. The FBG transverse and longitudinal geometries are also modelled, allowing us to calculate the coupling coefficients between the fibre modes. These coupling coefficients are then applied to the coupled mode equations, which are then solved over the length of the FBG for different FBG transverse locations within the fibre core. The meaning and theoretical composition of each step of the simulation is explained in the sections that follow.

## 3. Numerical Study Theory

### 3.1. Output Fibre Specifications and Modelling

The output fibre is a 10 m Prysmian graded index Low Differential Mode Group Delay (DMGD) 6 LP mode fibre. If one considers the mode degenerate states, the fibre contains 10 LP modes. Because the fibre is designed as a low DMDG fibre, the profile is designed such that the differences in effective refractive indices between modes of the same mode group are minimal. As a result, the modes of each mode group propagate through the fibre with a similar delay, and thus arrive at their destination at similar instances in time.

The refractive index profile of the Prysmian fibre in question can be found in [26]. However, this profile is the ideal case, while in reality, the refractive index profile may vary between specific batches. Therefore, for accurate numerical modelling, the refractive index profile was obtained experimentally using an IFA-100 interferometric fibre analyser (LUNA Innovations, Blacksburg, VA, USA) operating at 633 nm. This specific instrument allows for the refractive index profile to be determined with an accuracy of ±0.0001 with a spatial resolution of 500 nm. This refractive index profile, for the specific batch of the fibre in our possession, was then used in all following numerical studies (Figure 4).

As can be seen in Figure 4, there are several notable differences between the experimentally obtained and the literature refractive index profiles. Despite these differences, however, we see strong agreement with the literature values for the mode groups refractive indices.

An important observation is the presence of the first leaky-mode group, close to the guidance condition $n_{\mathrm{co}}>n_{\mathrm{eff}}>n_{\mathrm{cl}}$, where $n_{\mathrm{co}}$ is the maximum refractive index value of the core, $n_{\mathrm{eff}}$ is the effective refractive index of the mode group, and $n_{\mathrm{cl}}$ is the cladding refractive index. Low DMDG fibres are often inscribed close to the cut-off wavelength for the designed number of modes to ensure minimal losses in the design-mode groups. The fibre also possesses a trench-assisted core to rapidly suppress the evanescent field component of the core guided modes. As a result, this means that the leaky-mode group transverse electric-field distributions are likely to have a large overlap with the fibre core. Therefore, the modes of the first leaky-mode group have been included in our numerical study.

### 3.2. Linearly Polarised Fibre Modes

To study the modal composition of the Prysmian 6LP mode fibre, we solve Maxwell’s equations following the assumption of weakly guided modes. This well-known simplification renders the vector wave equation into the scalar wave equation. The effective refractive index values of the modes can be computed using the multilayer approach set out by Vassallo [27]. This method separates the refractive index profile of the fibre core into multiple thin slices from the transverse centre of the core along its radius. The core profile is modelled as *N* layers with the width of the *i*th layer expressed as $a_{i}$. The *N*th layer represents the beginning of the fibre cladding. At each interface between layers, the field continuity is numerically evaluated for effective refractive index values fitting the definition of a core guided mode, $n_{\mathrm{cl}}<n_{\mathrm{eff}}<n_{\mathrm{co}}$. Along the radius of the fibre, the field components contain a divergent component dependent on the modal propagation constant, with an amplitude *D*. If $D(\beta_{ij})=0$ at the core cladding interface, then the mode is guided by definition. This is due to the fact that the mode field distribution after the core-cladding interface is expressed solely by the modified Bessel function of the second kind, $K_{l}$, and thus converges to zero. Pre-normalisation, the LP mode transverse electric field profile is defined as, (1)$$E_{lm}\left(r,\theta \right)=\left\{\begin{array}{ccc} AJ_{l}\left(u_{i}\frac{r}{a}\right)+BY_{l}\left(u_{i}\frac{r}{a}\right)\left\{\begin{array}{c} \cos(l\phi ),\\ \sin(l\phi ) \end{array}\right.\; & \; & \mathrm{if}\;n_{\mathrm{eff}}<n_{i}\\ CI_{l}\left(w_{i}\frac{r}{a}\right)+DK_{l}\left(w_{i}\frac{r}{a}\right)\left\{\begin{array}{c} \cos(l\theta ),\\ \sin(l\theta ) \end{array}\right.\; & \; & \mathrm{if}\;n_{\mathrm{eff}}>n_{i} \end{array}\right.$$
where *A*, *B*, *C*, and *D* are amplitudes, $J_{l}$ and $Y_{l}$ are the Bessel functions of the first and second kind, $I_{l}$ and $K_{l}$ are the modified Bessel functions of the first and second kind, *a* is the radius of the fibre core, $u_{i}$ and $w_{i}$ are the scaler mode parameters following $u=ak_{0}\sqrt{n_{i}^{2}-n_{\mathrm{eff}}^{2}}$ and $w=ak_{0}\sqrt{n_{\mathrm{eff}}^{2}-n_{i}^{2}}$, with $k_{0}=2\pi /\lambda$ being the free-space wavenumber, and $n_{i}$ the refractive index of the *i*thlayer of the core. Applying this multilayer methodology to our fibre, we arrive at Figure 5 for the core guided modes. The modes can be sorted into groups following the equation g=2m+l−1, in which *l* and *m* are the azimuthal and radial mode orders, respectively. The effective refractive indices of these groups have been included in Figure 4.

Furthermore, as alluded to in the study carried out by Thomas et al. [9,10], localised FBGs possess the ability to couple core guided modes to the fibres many cladding modes. An important feature of the low DMDG fibre, as previously alluded to, is that the ensemble of the first leaky-mode group has effective refractive indices close to, yet below, that of the cladding. In the case of leaky modes, the electric field distribution does not become evanescent at the core cladding interface and oscillated radially over the cladding. As a result, while the first leaky-mode group is not guided by the core, the electric field distribution will have a non-negligible amplitude that remains within the core itself (Figure 6). Therefore, it is technically possible to transfer large quantities of energy to this leaky-mode group with an FBG. Nevertheless, the oscillating part of the leaky-mode electric field, situated in the fibre cladding, results in a rapid loss of energy as it is quickly transferred to the fibre coating. Therefore, any light coupled to this group will not be seen in the reflection spectrum. Nevertheless, modes coupling energy to this mode group will produce notable features in the FBG transmission spectrum.

### 3.3. MUX Hermitian Gauss Modes

To better understand how the HG modes excite the core guided LP modes, it is necessary to model them. HG modes are a set of possible solutions to the paraxial wave equation in Cartesian coordinates, coming together to form a mutually orthogonal set. These modes can be easily recognised by their rectangular symmetry in the transverse direction with their electric field distributions described by [28],(2)$${\mathrm{HG}}_{lm}\left(x,y\right)=C_{lm}H_{m}\left(\frac{\sqrt{2}}{w_{0}}x\right)H_{m}\left(\frac{\sqrt{2}}{w_{0}}y\right)e^{-\left(x^{2}+y^{2}\right)/w_{0}^{2}},$$
where $C_{lm}$ is the mode field normalisation factor, $H_{l}$ and $H_{m}$ are the Hermite Polynomials of orders *l* and *m*, respectively, and $2w_{0}$ is the beam waist. Each ${\mathrm{HG}}_{lm}$ mode of this set is defined by a pair of indices *l* and *m*, which refer to the number of nodes along the *x*- and *y*-axis, respectively. In the case of l=m=0, i.e., ${\mathrm{HG}}_{00}$, we are left with the fundamental mode, taking the form of a simple Gaussian beam. The formalism in Equation (Equation 2) was then applied in our simulations to model the Proteus output HG modes (Figure 7). It is important to note that at this stage, an arbitrary beam waist was used prior to its numerical estimation.

Although Figure 7 reproduces well the HG modes, questions are posed about their form when applied to our system. The HG modes are formed in free space within the Proteus; however, they are then coupled to the output fibre. This transition from free space to the circular geometry of the fibre should cause the form of the modes to distort away from their characteristic rectangular shape. Thus, a rudimentary validation was carried out.

### 3.4. HG Model Experimental Verification

To verify this representation of the HG modes, a simple setup was constructed. An infrared camera was used to take the far-field images of each of the output modes. The camera was focused on the cleaved end of the 10 m Proteus output fibre. The input modes were then excited in turn and a photo of each field intensity distribution taken (Figure 8).

As can be rapidly deduced from Figure 8, the output modes do not display the typical rectangular geometry one would expect to see from HG modes. Additionally, it is evident that the modes do not go through a complete transition to the circular geometry of the fibre, otherwise we would observe modes with Laguerre Gaussian semblance. In fact, the modes appear elliptical.

Furthermore, we note that the high-order mode intensity distributions appear to contain unexpected features, hinting at mode-mixing effects. The degradation from the sharp-mode fields we see in the case of ${\mathrm{HG}}_{00}$, ${\mathrm{HG}}_{01}$, and ${\mathrm{HG}}_{10}$, to the more chaotic high-order mode fields is most likely due to inter-modal coupling within the fibre LP mode groups. The nature of the Prysmian fibre used in this study dictates that the effective refractive indices between the mode groups are well separated; the contrary can be said within the mode group itself. Therefore, slight perturbations to the optical fibre, in the form of bending, for example, [29], may result in notable energy transfer between modes that occupy the same mode group and notably between modal degenercies [30]. Additionally, as the order of the mode increases and the field distribution becomes more complex, perturbations causing field distortions can easily result in non-zero overlaps of these mode fields.

To support this claim, it is prudent to highlight a clear case of mode mixing observable in Figure 8. The most easily interpretable case of mode mixing can be seen in the ${\mathrm{HG}}_{11}$ intensity distribution. This specific mode should theoretically excite group 3 LP modes only. Furthermore, the excitation modes ${\mathrm{HG}}_{02}$ and ${\mathrm{HG}}_{20}$ are the only other two excitation modes that should, under normal working conditions, also couple to the fibre via the group 3 LP modes. Here, it becomes evident that instead of the appearance of four maxima for HG11, as seen in Figure 7, we obtain a clear mix of the ${\mathrm{HG}}_{11}$, ${\mathrm{HG}}_{02}$, and ${\mathrm{HG}}_{20}$ mode fields, hinting at a distributed excitation across the group 3 LP modes.

Another possible cause of the mode field distortion may be due to the beat length between excited LP modes, and thus their vector mode composition, as the HG mode is coupled to the fibre. The beat length between two modes is defined as $L_{B}=2\pi /\Delta \beta$ [31,32]. Even though we selectively excite HG modes to the fibre, the excitation HG mode may couple to the fibre as a superposition of LP modes occupying the same group, as is the case for high-order modes. As a result, multiple modes of the same mode groups being excited at the same time will give rise to beating effects and potentially noticeable mode field distortion along the 10 m output fibre. For example, the approximate beat length between ${\mathrm{LP}}_{02}$ and ${\mathrm{LP}}_{21}$, both modes of group 3, is in the region of 5.4 cm at 1550 nm in this fibre.

### 3.5. Ince–Gauss Approximation

To better adapt the electric-field distributions of the HG modes, we opted to model the modes as Ince–Gauss (IG) modes. As taken from the extensive presentation of IG modes by Bandres and Gutiérrez-Vega [33], the IG modes are additional solutions to the paraxial wave equation, returning exact orthogonal solutions. While HG modes are solutions in Cartesian geometry, and LG modes solutions in circular geometry, the IG modes are solutions of the paraxial wave equation in elliptical geometry. These modes then represent the exact and continuous transition modes between HG and LG modes [33]. Here, we present the formalism outlined in [33] at a high level.

The IG modes are based on even and odd Ince polynomials in elliptical geometry. With the *z*-plane chosen as the transverse plane, the elliptical coordinates are defined as x=f(z)cosh(ξ)cos(η), y=f(z)sinh(ξ)sin(η) and z=z. Here, f(z) is the semi-focal separation with respect to *z*. Given $f\left(z\right)=f_{0}w\left(z\right)/w_{0}$ with $f_{0}$ being the semi-focal separation at z=0. ξ and η are the radial and angular elliptical variables, respectively, defined as ξ∈[0,∞] and η∈[0,2π]. The general expressions of the even and odd IG mode electric fields are as follows:(3)$$\begin{array}{c} IG_{lm}^{e}\left(r,\epsilon \right)=\frac{Cw_{0}}{w(z)}C_{l}^{m}\left(i\xi ,\epsilon \right)C_{l}^{m}\exp\left[\frac{-r^{2}}{w^{2}\left(z\right)}\right]\times \exp\left[i\left(kz+\frac{kr^{2}}{2R(z)}-\left(p+1\right)\psi_{GS}\left(z\right)\right)\right], \end{array}$$(4)$$\begin{array}{c} IG_{lm}^{o}\left(r,\epsilon \right)=\frac{Sw_{0}}{w(z)}S_{l}^{m}\left(i\xi ,\epsilon \right)S_{l}^{m}\left(\eta ,\epsilon \right)\exp\left[\frac{-r^{2}}{w^{2}\left(z\right)}\right]\times \exp\left[i\left(kz+\frac{kr^{2}}{2R(z)}-\left(p+1\right)\psi_{GS}\left(z\right)\right)\right]. \end{array}$$
where r denotes the radius, the ellipticity parameter is $\epsilon =2f_{0}^{2}/w_{0}^{2}$, C and S are normalisation constants, $C_{l}^{m}$ and $S_{l}^{m}$ are the even and odd Ince polynomials, respectively, of order *p* and degree *m*, k is the wave number, and $R\left(z\right)=z+z_{R}^{2}/z$ is the radius of curvature of the phase front, in which $z_{R}=kw_{0}^{2}/2$ is the Rayleigh range and $w_{0}$ the beam waist as z=0. Finally, the variable *p* is a separation constant and $\psi_{GS}$ the Guoy shift, given as $\psi_{GS}=arctan\left(z/z_{R}\right)$.

It is important to note that the orders and degrees of even and odd IG modes follow 0⩽m⩽l for even functions and 1⩽m⩽l for odd functions. Additionally, the parity of the indices l,m is always the same, i.e., ${\left(-1\right)}^{l-m}=1$.

The IG mode formalism is very versatile when used for modelling HG, LG, and IG modes alike. In fact, with this formalism, one can transition between the three mode field coordinate systems by defining the ellipticity constant. For example, LG modes are solutions to the paraxial wave equation in circular coordinates. Therefore, as $f_{0}\to 0$, in turn, ϵ→0, i.e, the distance between the two foci of the ellipse tends to zero, meaning the system tends towards spherical coordinates. The same can be said as $f_{0}\to \infty$, ϵ will also tend to infinity. This reformulates the system to Cartesian coordinates and HG modes.

The IG modes were then modelled using MATLAB R2023a (Figure 9). The ellipticity constant was chosen to be equal to 1.5 as this gave the best agreement for the modes that were not overly corrupted by model crosstalk in Figure 8, namely, ${\mathrm{HG}}_{02}$, ${\mathrm{HG}}_{20}$, ${\mathrm{HG}}_{03}$, and ${\mathrm{HG}}_{30}$. It is noted that ${\mathrm{HG}}_{00}$, ${\mathrm{HG}}_{01}$, and ${\mathrm{HG}}_{10}$ are identical to their LG and IG counterparts irrespective of coordinate geometry.

### 3.6. Free-Space to Fibre-Mode Coupling

When describing fibre modes, one tends to assume a perfectly cylindrical, weakly guided fibre; thus, the free-space HG modes are not possible fibre modes. Therefore, as the HG modes reach the interface between free space and the fibre, they are coupled to the fibre as a superposition of LP modes with differing complex coefficients. Hence, the incident HG mode can now be represented as,(5)$${\mathrm{HG}}_{lm}\left(x,y\right)=\sum_{i}^{\infty }\sum_{j}^{\infty }c_{ij}{\mathrm{LP}}_{ij}\left(x,y\right)+L_{lm},$$
where $c_{ij}$ is the complex coefficient for the given mode ij, with non-zero values eluding to excitation, ${\mathrm{LP}}_{ij}$ is the electric field distribution of the ijth linearly polarised mode, and $L_{lm}$ is a loss term added for completeness. A large $L_{ij}$ term implies imperfect coupling, where poor alignment or tailoring of the incident beam geometry may have resulted in leaky or cladding-mode excitation.

For each LP mode, the complex coefficient $c_{ij}$ is calculated as the overlap between itself and the incident HG mode,(6)$$c_{ij}=\underset{\infty }{\;\int \int \;}{\mathrm{HG}}_{lm}\left(x,y\right){\mathrm{LP}}_{ij}\left(x,y\right)dxdy.$$

Here the electric field distributions are normalised. Thus, the initial power in each LP mode upon excitation is given as,(7)$$p_{ij}={\left|c_{ij}\right|}^{2}.$$

### 3.7. Beam Waist Estimation

Finally, it was necessary to simulate the beam waist of the Proteus output at the moment of coupling to the fibre core. This value was not given, so the assumption was made that the beam waist would have been calculated to give optimum coupling of the HG mode to the fibre core. Therefore, it was designed to minimise the loss term, $L_{lm}$, in (Equation 5). Thus, the HG beam waist was chosen as the beam waist which resulted in the maximum coupling efficiency. The coupling efficiency is simply the sum of the excited optical powers,(8)$$\eta_{lm}=\sum_{i}^{\infty }\sum_{j}^{\infty }p_{ij}.$$

To find this optimal value, we calculated the overlap of a given HG mode with all LP modes for a wide range of beam waists (Figure 10).

Using this methodology, we are then able to calculate how the Proteus output HG modes fall into the four principal guided mode groups of the Prysmian 10 LP mode fibre, organised as in Table 1.

## 4. Localised FBG Modelling

### 4.1. FBG Refractive Index Modification Geometry

The manner in which we model the refractive index modification induced by the femtosecond laser follows the formalism laid out by Thomas, et al. [9,10]. In their work, they describe the refractive index modification in the transverse plane of the fibre as a homogeneous ellipse (Figure 11), this ellipse being the result of a type II modification. Although it has been observed that the type II modification is accompanied by a type I shell, this is omitted as the magnitude of the type I RIM contribution is several orders less than that of the type II modification zone. Referring back to Figure 2, we estimate the type II ellipse to have a height and width of h=2.2μm and w=0.8μm, respectively.

Along the length of the FBG, we assume the RIM to take the form of a periodic square modification. Naturally, the real refractive index modification in the y/z plane is identical to that of the x/y plane; a square modification is assumed for simplicity. Following slight modifications to the equations presented by Thomas et al. [10], the maximum refractive index modification in the transverse plane is defined as,(9)$$\Delta n_{\max}\left(x,y\right)=\left(n_{\mathrm{mod}}-n_{\mathrm{core}}\left(x,y\right)\right)\Theta \left(x,y\right),$$
where $n_{\mathrm{mod}}$ is the refractive index change caused by the laser, $n_{\mathrm{core}}\left(x,y\right)$ is the refractive index profile of the fibre core in the transverse plane, and Θ(x,y) is a function which defines the previously mentioned elliptical RIM geometry in the transverse plane. The term Θ(x,y) gives a value of zero or one, when outside or within the modification zone, respectively.

The grating-induced perturbation to the fibre core permittivity along its length can then be estimated as,(10)$$\Delta \varepsilon \left(x,y,z\right)\approx 2\varepsilon_{0}n\left(x,y\right)\Delta n\left(x,y,z\right)=\varepsilon_{0}n_{\mathrm{core}}\left(x,y\right)\left(\Delta n_{0}\left(x,y\right)+\sum_{v=1}^{\infty }\Delta n_{v}\left(x,y\right)2\cos\left(\frac{2\pi v}{\Lambda }z\right)\right).$$

Here, $\varepsilon_{0}$ is the permitivity of free space, Δn(x,y,z) is the refractive index change of the FBG, $\Delta n_{0}\left(x,y\right)$ is the d.c. refractive index change induced by the FBG, and $\Delta n_{v}\left(x,y\right)$ is the a.c. refractive index change with reference to the different Fourier orders *v*. In our case, we use only third-order FBGs, i.e., v=3.

The a.c. and d.c. portions of the refractive index changes can be further expanded as,(11a)$$\begin{array}{cc} \Delta n_{0}\left(x,y\right) & =\frac{2w}{\Lambda }\Delta n_{\max}\left(x,y\right), \end{array}$$(11b)$$\begin{array}{cc} \Delta n_{v}\left(x,y\right) & =\frac{2}{\pi v}\sin\left(\pi v\frac{w}{\Lambda }\right)\Delta n_{\max}\left(x,y\right), \end{array}$$

### 4.2. Coupling Coefficients

In the theory of fibre Bragg gratings, the coupling coefficient equation is well known. However, in our study, a small modification is needed to ensure the localised nature of the FBG is accounted for. Thus, the modal coupling coefficient between two modes ij and lm can be expressed as [34],(12)$$k_{ij-lm}=\frac{\omega }{4}\underset{\Theta }{\;\int \int \;}\Delta \varepsilon \left(x,y,z\right)E_{ij}E_{lm}^{*}dxdy,$$
where ω, the angular frequency, follows the relation ω=2πc/λ. Here, the integral is limited to Θ(x,y), the pre-defined elliptical geometry of the laser RIM. The key significance of this modification is that the localised-elliptical RIM breaks the circular symmetry of the mode fields, no matter its location in the transverse plane of the fibre core. Therefore, while the guided core modes make an orthogonal set, the following orthogonality relation does not hold for localised FBGs:(13)$$\underset{\infty }{\;\int \int \;}E_{ij}\left(x,y\right)E_{lm}^{*}\left(x,y\right)dxdy=\delta_{il}\delta_{jm},$$
with $\delta_{il}$ and $\delta_{jm}$ being Kronecker deltas, equal to zero when the two suffixes do not equate, and one if they do. Therefore, the localised nature of the RIM allows for complex intermodal cross-coupling between the core guided modes, irrespective of their orders. As a result, the coupling coefficients must be recalculated for any change in the RIM position in the core, dy or dx, and for any adaptation to the RIM transverse geometry Θ(x,y).

Consequently, it is useful to define two new equations for the d.c. and a.c. contributions to the mode coupling. Separating the coupling coefficient into two coefficients allows for more simplicity when applying the coefficient to the coupled mode equations. From Equations (Equation 10)–(Equation 12),(14a)$$\begin{array}{cc} k_{ij-lm}^{d.c.} & =\frac{w}{\Lambda }\Pi_{\Theta }^{ij-lm}, \end{array}$$(14b)$$\begin{array}{cc} k_{ij-lm}^{a.c.} & =\frac{1}{\pi v}\sin\left(\pi v\frac{w}{\Lambda }\right)\Pi_{\Theta }^{ij-lm}, \end{array}$$

Here, the summation of the Fourier components in the a.c. refractive index modification is dropped as only the FBG inscription order is of importance within the spectral window of the FBG, with $\Pi_{\Theta }^{ij-lm}$ representing the mode overlap integral within the void cross-section,(15)$$\Pi_{\Theta }^{ij-lm}=\frac{\varepsilon_{0}\omega }{2}\underset{\Theta }{\;\int \int \;}\Delta n_{max}\left(x,y\right)E_{ij}\left(x,y\right)E_{lm}^{*}\left(x,y\right)dxdy.$$

Once the fibre parameters and thus the mode field parameters are known, evaluating this integral becomes a powerful tool to predict RIM transverse locations of interest. This is especially useful for predicting locations for optimal energy exchange between specific modes for fibre lasers or mode converters, for example, without the need to fully simulate the FBG spectra, which can become time-consuming when evaluating fibres with large numbers of guided modes.

### 4.3. Spectra Realisation

The equations in ([Disp-formula FD14a-sensors-25-06087]) can then be used in conjunction with the coupled mode equations to obtain the FBG spectral response in both transmission and reflection. The coupled mode equations allow us to track the development of the modal amplitudes of the forward- and backwards-propagating modes along the length of the FBG [34,35]. The set of differential equations can be simplified to give the energy exchange between a mode ij and the complete set of modes in the fibre lm,(16a)$$\begin{array}{cc} \frac{dA_{ij}}{dz} & =i\left[k_{ij-ij}^{d.c.}A_{ij}+\sum_{l}\sum_{m}k_{ij-lm}^{a.c.}B_{lm}e^{-i(\beta_{ij}+\beta_{lm}-K)z}\right], \end{array}$$(16b)$$\begin{array}{cc} \frac{dB_{ij}}{dz} & =-i\left[k_{ij-ij}^{d.c.}B_{ij}+\sum_{l}\sum_{m}k_{ij-lm}^{a.c.}A_{lm}e^{i(\beta_{ij}+\beta_{lm}-K)z}\right], \end{array}$$
where *A* and *B* are the amplitudes of the transverse mode fields of the given mode ij in the forward and backward directions, respectively, β is the modal propagation constant, K=2πv/Λ is the grating wave vector, and the summation of the lm modes includes lm=ij. To obtain the complete reflection spectra, encompassing all modal cross-coupling, we expand the above equations so that ij represents each mode in the fibre consecutively, resulting in a set of 20 differential equations, 30 with the inclusion of the leaky modes.

These equations can then be solved simultaneously as an initial value problem, following the premise that $A_{ij}\left(0\right)=\sqrt{p_{ij}}$ and $B_{ij}\left(L\right)=0$. Here, the value $p_{ij}$ is calculated using Equation (Equation 7). This idea presumes that no energy is exchanged between the modes as the light traverses the fibre between the moment of being coupled to the fibre by the Proteus and reaching the beginning of the FBG. Although this assumption is somewhat naive, random intermodal mixing is impossible to predict and is thus omitted from our calculations. Nevertheless, the nature of the low DMGD fibre means that the majority of the random intermodal mixing occurs within the mode groups themselves.

### 4.4. Features of the Few-Mode Localised FBG Spectrum

To better understand the results obtained from both the numerical study and experimental validation, it is important to comprehend the origins of the features of the localised FM-FBG spectra. To do this, it is prudent to revisit the phase matching condition for a uniform FBG. Below are two equivalent expressions of the phase matching condition in the presence of multiple modes,(17a)$$\begin{array}{cc} \lambda_{B}^{ij-lm}= & \left(n_{\mathrm{eff}}^{ij}+n_{\mathrm{eff}}^{lm}\right)\frac{\Lambda }{v}, \end{array}$$(17b)$$\begin{array}{cc} \beta^{ij}= & -\beta^{lm}+\frac{2\pi v}{\Lambda }. \end{array}$$

Here, $\lambda_{B}^{ij-lm}$ then represents the location of the Bragg peak due to coupling effects between the forward-propagating ijth mode with the backward-propagating lmth mode. In the case ij=lm, the equation then naturally simplifies to the well-known FBG equation $\lambda_{B}=2n_{\mathrm{eff}}\Lambda /v$. Additionally, $\beta =2\pi n_{\mathrm{eff}}/\lambda$ is the modal propagation constant. It is important to note that the sign of β is directionally dependent and is negative since $-\beta^{lm}$ is the counter-propagating $lm^{th}$ mode.

The phase matching conditions in ([Disp-formula FD17a-sensors-25-06087]) then imply that the spectra will consist of a large quantity of peaks, the number of which is governed by the number of modes in the fibre. In our case, the Prysmian fibre contains 10 LP modes, including degeneracies. Therefore, each mode is capable of one self-coupling event and nine cross-coupling events. If we then consider all 10 guided modes, this gives 100 peak contributions to the spectrum.

However, due to the fact that the effective refractive indices of modes within the same groups are almost indistinguishable and that the effective refractive index difference between adjacent mode groups is well regimented at ∼$3.2\times 10^{-3}$, the multitude of contributions manifest as a handful of peaks containing the superposition of the contributions.

To better visualise this idea, it is useful to consider the phase matching condition in the propagation constant form, Equation ([Disp-formula FD17b-sensors-25-06087]) [35]. By plotting the propagation constants with respect to λ and noting the intersects with πv/λ, we recover the resonance locations (Figure 12).

Omitting leaky-mode coupling events and mode degeneracies for simplicity of presentation, the makeup of the mode group and cross-coupling resonance locations is as follows:Self-coupling resonance compositions:-$G_{1}∋\left\{{\mathrm{LP}}_{01}↔{\mathrm{LP}}_{01}\right\}$-$G_{2}∋\left\{{\mathrm{LP}}_{11}↔{\mathrm{LP}}_{11}\right\}$-$G_{3}∋\left\{{\mathrm{LP}}_{21}↔{\mathrm{LP}}_{21},\;{\mathrm{LP}}_{02}↔{\mathrm{LP}}_{02}\right\}$-$G_{4}∋\left\{{\mathrm{LP}}_{31}↔{\mathrm{LP}}_{31},\;{\mathrm{LP}}_{12}↔{\mathrm{LP}}_{12}\right\}$Cross-coupling resonance compositions:-$C_{1}∋\left\{{\mathrm{LP}}_{01}↔{\mathrm{LP}}_{11}\right\}$-$C_{2}∋\left\{{\mathrm{LP}}_{01}↔{\mathrm{LP}}_{21},\;{\mathrm{LP}}_{01}↔{\mathrm{LP}}_{02}\right\}$-$C_{3}∋\left\{{\mathrm{LP}}_{01}↔{\mathrm{LP}}_{31},\;{\mathrm{LP}}_{01}↔{\mathrm{LP}}_{12},\;{\mathrm{LP}}_{11}↔{\mathrm{LP}}_{21},\;{\mathrm{LP}}_{11}↔{\mathrm{LP}}_{12}\right\}$-$C_{4}∋\left\{{\mathrm{LP}}_{11}↔{\mathrm{LP}}_{31},\;{\mathrm{LP}}_{11}↔{\mathrm{LP}}_{12},\;{\mathrm{LP}}_{02}↔{\mathrm{LP}}_{21}\right\}$-$C_{5}∋\left\{{\mathrm{LP}}_{21}↔{\mathrm{LP}}_{31},\;{\mathrm{LP}}_{21}↔{\mathrm{LP}}_{12},\;{\mathrm{LP}}_{02}↔{\mathrm{LP}}_{31},\;{\mathrm{LP}}_{02}↔{\mathrm{LP}}_{12}\right\}$-$C_{6}∋\left\{{\mathrm{LP}}_{31}↔{\mathrm{LP}}_{12}\right\}$

It is important to note that the left–right arrow denotes the fact the coupling is bidirectional. For example, the resonance location C1 is a superposition of both ${\mathrm{LP}}_{01}\to {\mathrm{LP}}_{11}$ and ${\mathrm{LP}}_{11}\to {\mathrm{LP}}_{01}$, where the magnitude of each contribution to the resonance amplitude is dictated by the energy in each mode at the instance of coupling. In contrast, the coupling coefficient is identical in both cases. Furthermore, the cross-coupling groups C1 to C6 are located at a wavelength directly between the two groups between which energy is being transferred, i.e., $\lambda_{C_{ij}}=\left(\lambda_{G_{i}}+\lambda_{G_{j}}\right)/2$.

To depict this, we now refer our attention to the simulated example of an FBG with a transverse eccentricity of dx=0μm (Figure 13). This figure represents a theoretical case where all core guided modes are excited evenly. In other words, each mode contains 10% of the power coupled to the fibre. This is done to try and replicate an FBG excited under standard circumstances, meaning without the use of a spatial mode multiplexer to excite the FBG. An equivalent would be to remove the spatial mode multiplexer in Figure 1 and directly connect the FMF, containing the FBG, to the circulator.

We see that when the FBG is inscribed centrically within the fibre, coupling is only possible for modes with non-zero electric fields at the transverse centre of the core. This limits the core guided mode resonance locations to G1, G3, and C2. The coupling events that comprise these resonances are ${\mathrm{LP}}_{01}\Leftrightarrow {\mathrm{LP}}_{01}$, ${\mathrm{LP}}_{02}\Leftrightarrow {\mathrm{LP}}_{02}$, and ${\mathrm{LP}}_{01}\Leftrightarrow {\mathrm{LP}}_{02}$. This greatly simplifies the spectral composition from seven possible peaks to three. However, we note the appearance of a peak in the transmission spectrum in the G4/C6 location. Furthermore, this peak does not appear in the reflection spectrum. This is because this peak is the product of core-to-leaky mode coupling, namely, ${\mathrm{LP}}_{02}\to {\mathrm{LP}}_{03}$. As ${\mathrm{LP}}_{03}$ is not core-guided, it rapidly dissipates energy and would therefore not be present in the reflection spectrum. Additionally, upon closer observation of the peak at G3/C4, it is noticable that the reflection spectrum is not complementary to the transmission spectrum. This is for the same reason as above, as ${\mathrm{LP}}_{01}\to {\mathrm{LP}}_{03}$ cross-coupling causes a larger transmission dip, whereas the reflection spectrum only contains the ${\mathrm{LP}}_{02}\Leftrightarrow {\mathrm{LP}}_{02}$ self-coupling contribution.

Moving our attention to example spectra of FBGs with transverse eccentricities of dx=2μm and dx=6μm Figure 14, excited in the same way as Figure 13, it becomes obvious that the spectra become difficult to interpret. The act of adding a slight eccentricity means that all modes have non-zero electric field components which overlap within the RIM region, causing a large number of cross-coupling contributions.

Therefore, it is evident that simply applying a localised FBG to an FMF, without control of the modes initially excited in the fibre, provides limited control over the spectrum. The extent of this control can be split into three regimes for the localised FBG eccentricity: centric, low eccentricity, and high eccentricity. Here, the three regimes can be expressed as dx=0μm, 0μm <dx<4μm, and 4μm < dx < 10 μm, respectively, for our fibre.

In the low-eccentricity regime the electric field distributions for modes of all mode groups are important around the RIM location. This is the regime depicted by the red line of Figure 14, in which nearly all mode groups contribute to self-coupling and cross-coupling spectral features. As a result, we obtain a full and relatively complex spectrum.

If we inscribe the FBG into the high-eccentricity region, the black line of Figure 14, we begin to see the onset of a relative simplification of the spectrum. In this region, the amplitudes of the electric field distributions for the low-order mode groups begin to drop off rapidly. Thus, as the eccentricity increases within this region, the contributions of the low-order modes tend to zero. Spectrally, this causes the amplitudes of the resonant peaks associated with these modes to fall with eccentricity. In fact, in cases where dx>5μm, ${\mathrm{LP}}_{01}$ no longer contributes to the spectrum, entirely removing the G1 resonance position from the spectrum. Therefore, in this region, high-order mode cross-coupling dominates. As a direct result, this region can be used for high-order mode conversion, or filtering of high-order mode components from the output signal.

However, it is important to note that the high-order mode energy is distributed over a large area within the core, not localised to a Gaussian over the fibre transverse centre such as ${\mathrm{LP}}_{01}$. As a result, it is difficult to achieve a strong coupling with the localised FBG. The relatively low profile of the RIM means that modal overlap over its transverse cross-section tends to be small. Therefore, strong or very long gratings are needed for high reflectivity of the high-order modes. To visualise this phenomenon, the transmission and reflection spectra in Figure 14 have been normalised with respect to the maximum transmission loss and maximum reflectivity of the centric case presented in Figure 13.

As a result, the spectral simplifications possible with localised FBGs are not entirely sufficient to adapt the FM-FBG for use in sensing. This is due to the fact that the spectrum remains complex over a large bandwidth, limiting its usability for WDM. Nevertheless, these mode-conversion effects may prove useful for applications such as fibre lasers, especially for applications involving large mode-area fibres. To cite several examples, highly eccentric FBGs could act to filter out high-order modes, or centric FBGs can be used to convert ${\mathrm{LP}}_{01}$ to ${\mathrm{LP}}_{02}$. Therefore, while this has its limitations for sensing, other applications may well be pertinent.

Finally, it is important to express the fact that the above results are simulated. In a real-world scenario, it is not possible to know exactly how much energy is coupled to each of the modes within the fibre, in the absence of selective excitation. This example, with all modes excited evenly, is then not entirely physical. In a real scenario, the coupler would project the ${\mathrm{LP}}_{01}$ mode onto the Prysmian fibre. The effective area of this mode, depending on the fibre in the coupler, would then dictate the superposition of modes to be excited in the Prysmian fibre. Furthermore, the need to use a rapid connected coupler evokes misalignment between the coupler and Prysmian fibre, further confusing the distribution of excited modes. In addition to this, the reflection spectrum is unobtainable. This is due to multimode interference effects excited as the reflected signal passes from FMF to SMF at the coupler. Therefore, validation of our model cannot be fully expressed here.

However, a brief validation can indeed be carried out. Regardless of the modal power distribution excited in the FMF, the simplifications seen in the transmission spectrum of Figure 13 are ever possible. While peak amplitudes may vary, the absence of peaks in the C1, C3, and C5 locations is inherent to the centric FBG if our model is indeed correct. This effect can be seen in Figure 15.

## 5. Results and Discussion

### 5.1. Validation of the Numerical Model

Before making a direct comparison between the numerical and experimental results, it is first important to explain one of the most striking spectral effects caused by the use of a spatial mode multiplexer. Here, we are referring to the reflection spectrum, examples of which can be seen in Figure 16 and Figure 17. As the reflected light re-passes over the spatial mode multiplexer before reaching the CT400, the modal contributions from the reflection are sorted within the Proteus and leave through their designated SMF-28 excitation fibre. This is due to the fact that the MPLC technology used within the Proteus allows it to be used as both a MUX and DEMUX [18]. As a result, the reflection spectrum contains only the LP mode contributions that have a non-zero overlap with the excitation mode. This in itself is a very important result as this immense spectral simplification opens the door for WDM of FM and MM-FBGs.

To confirm the validity of our numerical model, several FBGs were inscribed with eccentricities of 0μm ⩽dx⩽6μm at 1μm intervals with lengths of 5 mm. Eccentricities higher than these values did not produce strong enough reflections to overcome the losses involved when re-traversing the Proteus. To maintain a standard point of reference, no eccentricity was intentionally applied along the y-axis; nevertheless, a slight parasitic eccentricity, dy, is unavoidable in reality. We then simulate the same conditions using our Matlab code and compare the experimental and numerical data. We arbitrarily apply a modulation depth in our code so that the reflection spectral peak is in relative agreement between the experimental and simulated results.

In our study, we obtain the best agreement between the numerical and simulated spectra when working with ${\mathrm{HG}}_{00}$ excitation, as can be seen in Figure 16, where ${\mathrm{HG}}_{00}$ excitation has been combined with a centric FBG. The amplitudes are in relatively good agreement in terms of amplitude. Some discrepancy does arise when we observe the wavelength position of the resonant peaks. We believe this is due to the effective refractive indices of the modes being calculated with fibre refractive index values at 1550 nm. The refractive index decrease at smaller wavelengths was not included in the calculation, hence why our resonances appear at higher wavelength values. Nevertheless, this does not affect the validity of our simulations.

Additionally, while not highlighted in Figure 13 and Figure 14, as mentioned previously, the localised nature of the FBG allows coupling to the first leaky-mode group. Each of these mode groups will then add additional contributions to the spectrum as they exchange energy to the leaky-mode group. This phenomenon can be directly observed in Figure 16. As ${\mathrm{HG}}_{00}$ excitation manifests itself within the fibre as pure ${\mathrm{LP}}_{01}$ excitation, the spectral features are solely due to ${\mathrm{LP}}_{01}$ self- and cross-coupling. Furthermore, as the FBG is centric, the only cross-coupling possible with the set of core modes present within the fibre is due to ${\mathrm{LP}}_{01}\Leftrightarrow {\mathrm{LP}}_{02}$, meaning, core mode coupling only accounts for two of the peaks present in the transmission spectrum. The peak located at the lowest wavelength is the result of ${\mathrm{LP}}_{01}$ coupling to the first group of leaky modes, namely, ${\mathrm{LP}}_{01}\Leftrightarrow {\mathrm{LP}}_{03}$, hence the inclusion of the leaky-mode group to our numerical assessment.

Discrepancies begin to arise when higher-order modes are numerically investigated (Figure 17). These discrepancies are seen as differences in amplitudes in the transmission and reflection spectra. We believe these differences arise due to the random inter-modal mixing between modes within mode groups as previously discussed. Furthermore, the higher-order HG modes excite a superposition of LP modes within the same mode group. Therefore, as the reflection spectrum is filtered by the Proteus, parts of the reflected power are sorted to a mode other than the excitation mode, leading to a reduction in the reflection peak amplitude.

While these discrepancies seem considerable, in reality, our numerical investigation does an excellent job at predicting the appearances of resonances and their relative amplitudes, when compared with each other. Additionally, the transmission spectra, despite differences in amplitude, clearly depict the same modal energy-exchange dynamics between modes and their mode groups. Therefore, our simulation remains a useful tool for predicting mode conversion effects with respect to FBG transverse location as well as the possibilities for spectral simplification. Therefore, while there are amplitude differences, our numerical results can be thought of as the best-case scenario.

One final spectral feature of note is the low-amplitude peaks seen in the reflection spectra of both Figure 16 and Figure 17. These smaller peaks, located at lower wavelengths in these specific examples, should not be present under perfect circumstances, as the modes form an orthogonal set. As a result, only one peak should be present in the reflection spectrum for any given modal excitation. To maintain agreement between the experimental and simulation results, we forced the appearance of these secondary peaks by assuming imperfect alignment between the Proteus output beam and the fibre core. The sub-peaks present in these two examples appeared after a transverse misalignment of dy=dx=1μm was applied, a fair assumption even for high-precision instruments such as this.

### 5.2. Transmission Spectral Control

Due to the extensive number of possible configurations of eccentricity and excitation mode, it is prudent to rely on the numerical model to save time. The transmission density plot highlighted in Figure 18 presents the extent of spectral control possible when observing the fibre transmission, the extent of which is limited, with the minimum number of transmission resonant dips obtainable being three.

While the transmission spectra do not appear to undergo great simplifications under single-mode excitation, subtleties in the spectra display interesting features. Namely, the maxima and minima of the transmission density plot allude to eccentricities which allow for strong control over the modal coupling effects. Furthermore, as each transmission density plot is obtained by single-mode excitation, all the resonance dips, except those related to self-coupling, represent uni-directional coupling events. Therefore, the plots paint a picture as to how the energy is converted following interaction with the inscribed FBG.

Several key examples in Figure 18 would be the eccentricities of dx=0μm, dx=4μm, and dx>6μm. Starting with the centric case of dx=0μm, we see that inscribing the FBG in this location means that no energy is transferred to G2 or G4. This can be seen as the absence of C1 for ${\mathrm{HG}}_{00}$ excitation, C3 and C5 for ${\mathrm{HG}}_{02}$ excitation, and the lack of peaks in the ${\mathrm{HG}}_{01}$ and ${\mathrm{HG}}_{21}$ plots.

In all three cases dx=0μm, dx=4μm, and dx>6μm, we find transverse inscription locations in which modes, and even entire mode groups, can pass over an FBG unperturbed. This is visualised in Figure 18 as ${\mathrm{HG}}_{00}$, ${\mathrm{HG}}_{01}$, and ${\mathrm{HG}}_{02}$ produce spectral responses at dx=4μm, for example. However, at this eccentricity, ${\mathrm{HG}}_{21}$ passes over the FBG without any interaction. While only ${\mathrm{HG}}_{21}$ is presented, this holds true for all other G4 excitation modes. A physical interpretation of this is that if a very strong grating is inscribed at this eccentricity, even when the initial modal composition coupled to the fibre is uncontrolled, the FBG would act to filter out the other mode groups at specific wavelengths, with G4 pass unperturbed.

Therefore, while the transmission spectrum is not adequately simplified for sensing applications, it becomes an important tool for understanding the energy-exchange dynamics caused by FBG eccentricity.

### 5.3. Wave Division Multiplexing

Referring once more to the transmission density plot shown in Figure 18, we can gather an understanding of the raw reflected signal, unfiltered by the Proteus. As is the case with type II FBGs, the reflection spectrum is expected to be somewhat complementary to the transmission spectrum. Thus, while we excite the fibre with one mode at a time, the reflected signal will contain a mix of several modes due to the intermodal cross-coupling forces by the FBG, both within mode groups and to any of the other mode groups. However, as previously stated, instead of recuperating the reflected signal with an optical circulator before the light re-traverses the Proteus, we recuperate the signals with a circulator placed before the SMF-28 excitation fibres. As a result, the reflected signal re-traverses the proteus, and the superposition of LP modes within the signal is sorted into its HG mode equivalence, the mechanism of which is the direct reverse of the phenomenon explained in Equation (Equation 5). We can see how this manifests in the reflection density plot of Figure 19.

This spectral filtering effect is interesting in itself, as it allows for WDM in a manner identical to that of an SM-FBG. If no spectral filtering was present, the large spectral range of the FM-FBG resonance peaks would mean that an interrogator, with a bandwidth in the C-band, could not even multiplex two FBGs in this manner, when inscribed to avoid resonant peak overlaps. This is then naturally inadequate for applications dependent on large arrays of sensors.

If one considers the case in which the FBGs undergoes WDM in a manner identical to that of a single-mode fibre, the transmission spectra would be chaotic due to excessive peak overlaps and would therefore be unusable in their raw state. However, if we then apply spectral filtering of the reflection spectrum, it would be possible to recover the spectral contributions of a single mode. As a result, each mode could be interrogated in turn via the Proteus, showing a standard WDM spectrum of an SM-FBG. Additionally, the capacity to separately interrogate multiple modes allows us to take multiple measures for each FBG, resulting in increased precision of measurement for sensing applications.

To perform this WDM with FM-FBGs, the use of whole core FBGs, i.e., those produced by interferometric inscription methods such as phase masks, would be more appropriate than localised FBGs. This is due to the fact that the void geometry parameter, Θ(x,y), in Equation (Equation 12) would equate the core transverse geometry. As a result, the modal coupling would once more be governed by the orthogonality relation, meaning only self-coupling would exist in the fibre. This means that less energy is lost because of unnecessary modal cross-coupling effects, resulting in stronger self-coupling resonances.

### 5.4. Spatial Mode Division Multiplexing

Furthermore, to the best of our knowledge, we believe that we have found a new way to multiplex FBGs by using selective modal excitation, spectral filtering, and FBG eccentricity. Referring to the ${\mathrm{HG}}_{00}$ and ${\mathrm{HG}}_{01}$ excitations in the reflection density plot of Figure 19, there are two zones of dx of high importance. First, if we consider the case in which dx=0μm, we see that ${\mathrm{HG}}_{00}$ will create a strong reflection due to ${\mathrm{LP}}_{01}\Leftrightarrow {\mathrm{LP}}_{01}$ self-coupling. However, at the same value of dx, the ${\mathrm{HG}}_{01}$ excitation produces no peaks. Thus, the ${\mathrm{LP}}_{11b}$ mode, excited in the fibre by ${\mathrm{HG}}_{01}$, traverses the length of the FBG without interaction. Therefore, even in the presence of multiple FBGs localised to the centre of the fibre core, the ${\mathrm{LP}}_{11b}$ mode is able to travel the length of the fibre unperturbed.

Moving the attention now to eccentricity values of dx>6μm, we observe a reversal of this situation. The ${\mathrm{LP}}_{01}$ mode will now pass over the FBG without interaction and the ${\mathrm{LP}}_{11b}$ mode will undergo coupling to a variety of modes propagating in the opposite direction. These resonance peaks, which can be seen in the transmission density spectrum, Figure 18, are then due to the ${\mathrm{LP}}_{11b}\Leftrightarrow {\mathrm{LP}}_{11b}$ self-coupling, along with ${\mathrm{LP}}_{11b}\Rightarrow {\mathrm{LP}}_{02}$, ${\mathrm{LP}}_{11b}\Rightarrow {\mathrm{LP}}_{12b}$, ${\mathrm{LP}}_{11b}\Rightarrow {\mathrm{LP}}_{31b}$ and ${\mathrm{LP}}_{11b}$ to several of the leaky modes. Once the eccentricity surpasses this value, the ${\mathrm{LP}}_{11b}\Rightarrow {\mathrm{LP}}_{01}$ coupling coefficient is negligible and the associated peak is no longer present; thus, almost no energy spills into the ${\mathrm{LP}}_{01}$ mode group.

Therefore, two FBGs possessing the same period could then be written into the same fibre. Inscribing one FBG central to the fibre core and the other with a high eccentricity would make the FBGs selective with regard to the modal composition of the excitation signals. This can therefore be thought of as a kind of FBG mode division multiplexing, which is, to the best of our knowledge, the first example of this phenomenon.

This new form of FBG multiplexing has a direct implication for FBG sensing arrays. If the concept is scaled up to arrays of FBGs, with their periods distinctly organised for WDM, one array can be inscribed centric, and the other highly eccentric. By doing so, we obtain a way to effectively double the number of FBGs in the WDM array.

In its standard single-mode FBG configuration, WDM is limited by the interrogator bandwidth and the wavelength separation between adjacent Bragg peaks. This ultimately couples together the dynamic range of each FBG and the maximum number of FBGs that can be included in an array. Thus, if one were to increase, the other must be decreased. A common method to increase the multiplexing capability of WDM arrays is to employ the use of Time Division Multiplexing (TDM). However, this method depends on complex synchronisation techniques to function correctly and is therefore strongly affected by any jitter or computer lag in the interrogation system.

Furthermore, the nature of the TDM method forces a minimum distance between FBGs of the same period to avoid signal overlap. This then impacts sensor spacing and may hinder the formation of dense quasi-distributed sensor arrays requiring a large number of sensors. Therefore, our presented method may provide an alternative to TDM for increasing the capacity of FBG sensing arrays distributed using the WDM method.

However, one obvious issue to highlight is the rotational dependence of each mode with respect to the FBG transverse location. An initial intuition would cause one to question the effectiveness of this method for multiplexed FBGs. Any slight twist applied to the fibre would cause the mode and RIM to become misaligned. Thus, FBGs inscribed separately and spliced together would naturally be rotationally misaligned with each other. Nevertheless, if we consider the overarching mode groups, and not the individual modes, we can see that the groups are rotationally independent. If the spectra are obtained individually and summed in post-process, we obtain an overall group contribution which is rotationally independent. This idea has been applied in Figure 19.

Additionally, it must be noted that at each FBG, energy spills into mode groups 3 and 4 due to modal cross-coupling. However, as this spillage occurs into the reflection spectrum and as the spillage occurs around a small resonance position at a wavelength shorter than the concerned Bragg resonances for groups 1 and 2, if the FBGs are multiplexed from longer to shorter wavelengths, this modal crosstalk should not be significant. Furthermore, back reflections from FBGs along the length of the multiplexed array would be expected to be minimal, as the modes would have to undergo multiple reflections before being recuperated by the interrogator. Therefore, if parasitic peaks do become present, they are expected to be small and easy to filter out with a cut-off reflectivity in the reflection spectrum.

### 5.5. Discussion on Localised FBG Optimisation

In this work, we present numerical and simulated results that highlight the capacity of this given set-up for WDM of FBGs in an FMF, as well as the introduction of a new type of modal multiplexing of FBGs in an FMF. Nevertheless, we present one specific case of both the FBG longitudinal and transverse geometry. Adaptations to these variables may help to optimise this proposed configuration for sensing applications. The most notable outcome is the increase in the FBG reflectivity to better suit commercially available interrogation systems.

We first consider the transverse geometry of the localised FBG. In this study, the RIM of the FBG is highly localised when compared to the relatively large size of the fibre core. Future studies may well consider decreasing the localisation by increasing the overall transverse area of the modification. Nevertheless, if the goal is to perform the modal multiplexing proposed in this work, great care must be taken in doing so. One proposition would be to increase the overall height of the elliptical RIM through beam shaping or simply by using an objective with a lower NA. If care is taken when positioning the FBG, making sure to keep the height of the ellipse symmetric over the x-axis, this may result in favourable reflection strengths. This increased reflectivity would be a direct result of an increased overlap integral and therefore coupling coefficient. Furthermore, the symmetry over the x-axis, along with the increased RIM height would begin to restore radial symmetry of the fibre core, suppressing modal cross-coupling effects. Nevertheless, over-extending the height of the RIM could cause centric FBGs to become visible to the ${\mathrm{LP}}_{11}$ mode degeneracy pair, meaning any parasitic y-offset would result in ${\mathrm{LP}}_{11}$ self-coupling, rendering modal multiplexing impossible.

To the opposite effect, one may consider decreasing the area of the transverse RIM geometry by using a higher NA objective. While at first glance this may appear to decrease the coupling coefficients, and thus reflectivity, it should result in other beneficial outcomes. Firstly, the increased localisation would result in less crosstalk. The RIM could be placed, with a higher localisation, within nodes of the modal electric field distributions. As a result, this would help to suppress any apparent crosstalk in the system. Furthermore, a resulting decreased width of the FBG would also cause the duty cycle of the FBG, in the longitudinal direction, to decrease. If this width is sufficiently tightened, it may become possible to inscribe second-order FBGs. The step towards second-order FBGs may result in an increase in reflectivity due to a 50% increase in the number of grating periods. However, an investigation should be carried out to compare the effects of decreasing the RIM geometry and moving from third- to second-order FBGs.

Lastly, a somewhat clear improvement that can be made to increase the FBG reflectivity is increasing the FBG length. In this study, we maintain a constant length of 5 mm for all FBGs. This was done so the FBGs could be produced rapidly and in a somewhat standard manner. However, as the maximum reflectivity of a Bragg peak can be estimated as $R=\tanh^{2}\left(kL\right)$ [34], one would expect increasing the length to have a very important effect on the usability of the setup studied in this work. Nevertheless, a study should be conducted to determine to what extent the length can be altered. Increasing the length to extreme amounts will result in excessive thinning of the Bragg peaks. This may result in the appearance of unwanted peak splitting.

## 6. Conclusions

We have presented and discussed each element of an extensive numerical model used to simulate localised FBGs in an FMF excited selectively via the use of a Cailabs Proteus-S spatial mode multiplexer. This study includes an experimental contribution that validates our simulation, while also highlighting its limitations.

We find that using a localised FBG without a spatial mode multiplexer leads to very limited simplifications. These simplifications may have practical uses in fibre lasers or communications; however, their applicability to sensing is limited.

By applying a spatial mode multiplexer to the FM-FBG, we find that the transmission spectrum remains complex. The use of single-mode excitation helps to visually express modal coupling effects across transverse locations but does not seem to have many practical applications.

Nevertheless, we find that by passing the reflected signal back over the spatial mode multiplexer, we can greatly simplify the reflection spectrum, stripping it back to its single-mode composition. This spectral simplification, coupled with logically choosing the FBG location, or with a whole-core FBG, allows for WDM of FM-FBGs.

Furthermore, we find that tailoring the FBG eccentricity to the excitation mode outputted by the spatial mode multiplexer can result in a new type of FBG multiplexing, to the best of our knowledge. By selecting the transverse location of the FBG, it is possible to dictate the individual modal coupling strengths between any given modes. We note that centric and highly centric FBGs provide locations in which specific modes do not interact at all with certain others. This allows for the production of FBGs that are selective towards the chosen excitation modes.

### Future Work

While we have theoretically shown that multiplexing via modal selection is possible, further studies should be developed to provide experimental validation.

Furthermore, the fibre and spatial mode multiplexer contains a large number of modes, as well as HG-LP mode conversion, adding unnecessary complexity to the results. It could be very interesting to investigate an FM-fibre with less supported modes with a spatial mode multiplexer that works directly in LP modes. This will allow for further refining of our numerical model under these simplified circumstances, and to see if a fibre that supports fewer modes will prove a better candidate for multiplexing via mode selectivity.

## Figures and Tables

**Figure 1 sensors-25-06087-f001:**
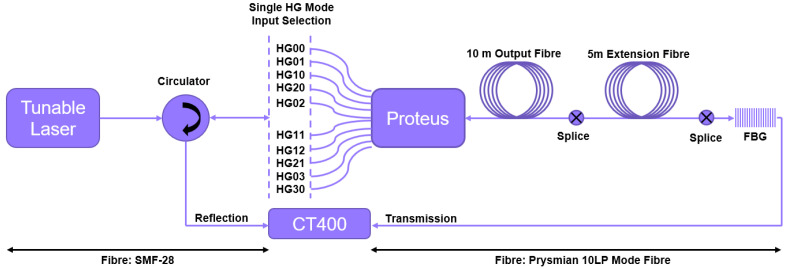
Experimental setup to excite modes selectively within a few-mode fibre containing an FBG. This setup also includes the method of interrogation in both the transmission and reflection.

**Figure 2 sensors-25-06087-f002:**
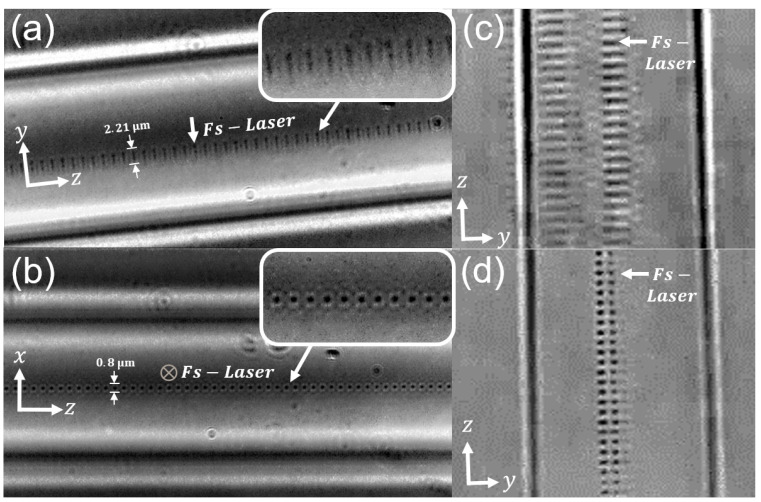
Images of FBGs written within the fibre core, showing (**a**) a side and (**b**) a top view of a localised FBG taken with a phase contrast microscope, bright-field images of (**c**) a double refractive index modification due to a high femtosecond pulse power and (**d**) multiple modifications forming due to re-passing the laser several times during inscription.

**Figure 3 sensors-25-06087-f003:**
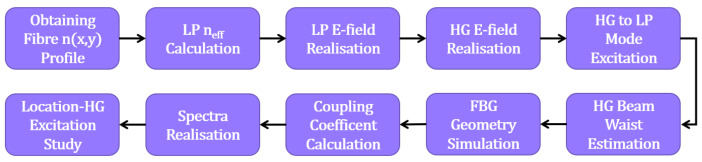
Flowchartof necessary steps to simulate the selective excitation of localised FBGs with a spatial mode multiplexer.

**Figure 4 sensors-25-06087-f004:**
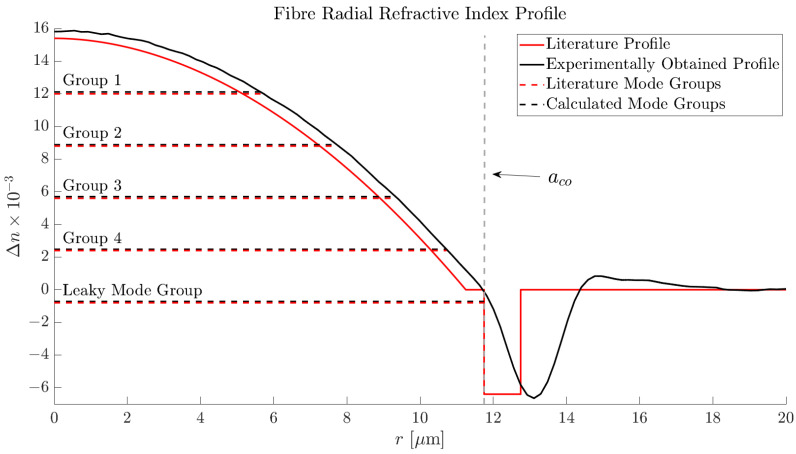
A comparison between the experimentally obtained and literature refractive index profile of the Prysmian fibre. Also highlighted are the calculated and literature mode group effective refractive indices as well as the core radius $a_{co}$.

**Figure 5 sensors-25-06087-f005:**
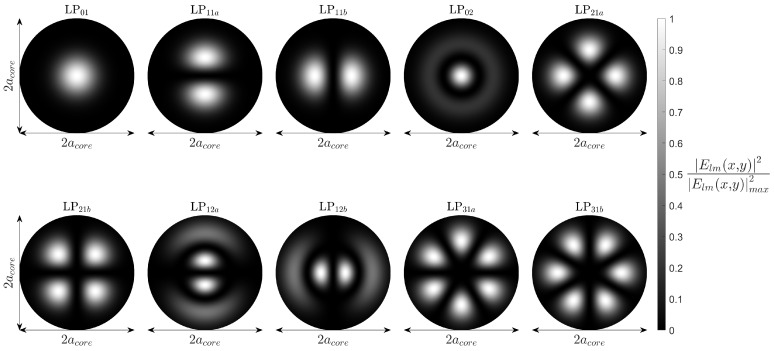
The simulated core guided mode electric field intensities of the Prysmian FMF, including degenerate states.

**Figure 6 sensors-25-06087-f006:**
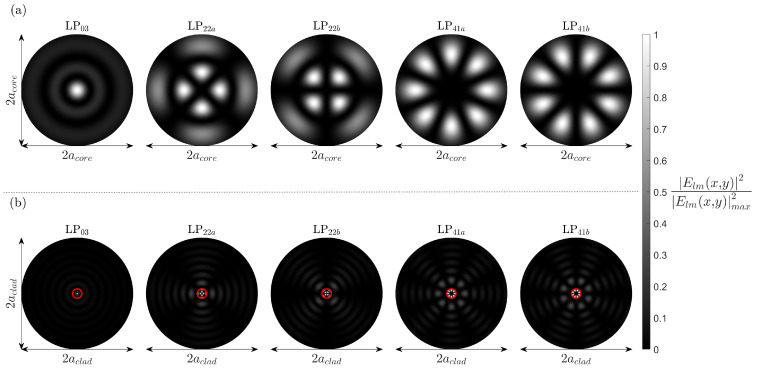
The simulated mode electric field intensities of the first leaky-mode group (**a**) in the core and (**b**) in the cladding of the Prysmian FMF, including degenerate states. The red circles found in (**b**) highlight the location of the core.

**Figure 7 sensors-25-06087-f007:**
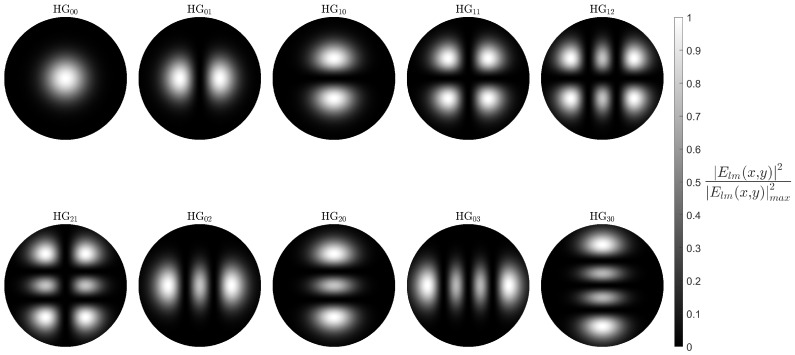
Simulations of the 10 HG modes produced by the Cailabs Proteus S. Here, the transverse *x* and *y* scale has been omitted as the beam waist has not yet been calculated and an arbitrary value used.

**Figure 8 sensors-25-06087-f008:**
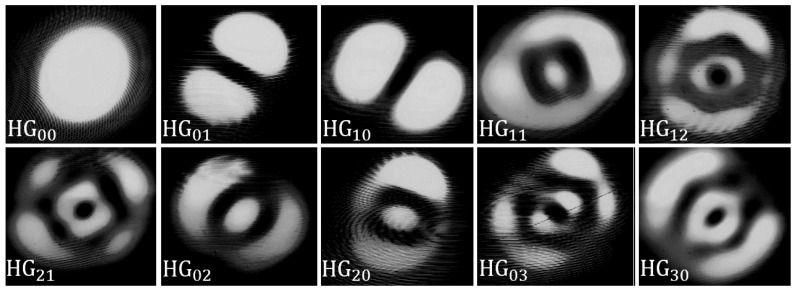
The far-field images taken of each HG mode at the cleaved end of the Proteus output fibre. Here, only one mode was selectively excited for each of the output photos.

**Figure 9 sensors-25-06087-f009:**
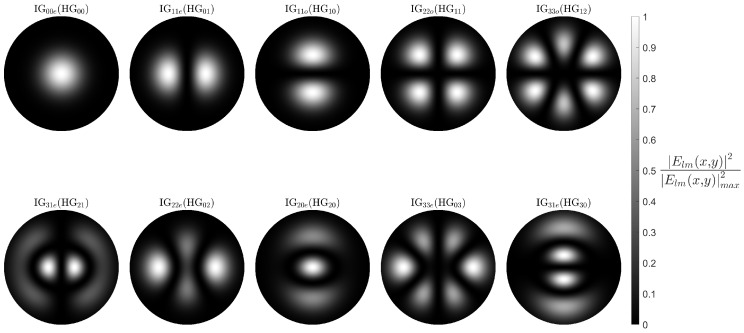
Simulated IG mode fields using an arbitrary beam waist. Each IG mode is noted along with the HG mode of origin before transitioning to elliptical geometry.

**Figure 10 sensors-25-06087-f010:**
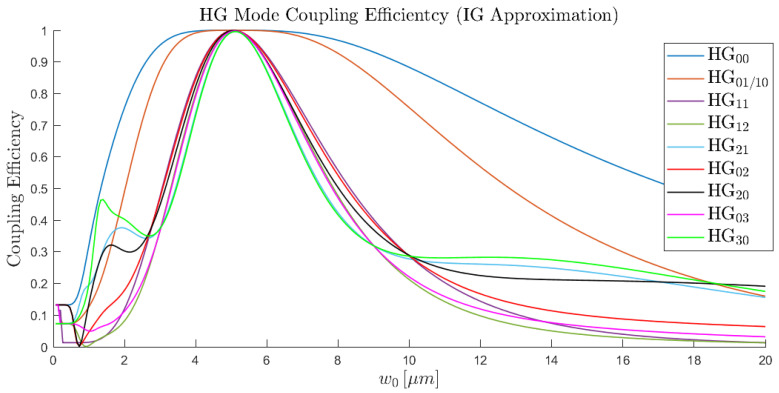
The beam waist approximation for each of the excitation HG modes using the IG mode approximation for the electric field distributions.

**Figure 11 sensors-25-06087-f011:**
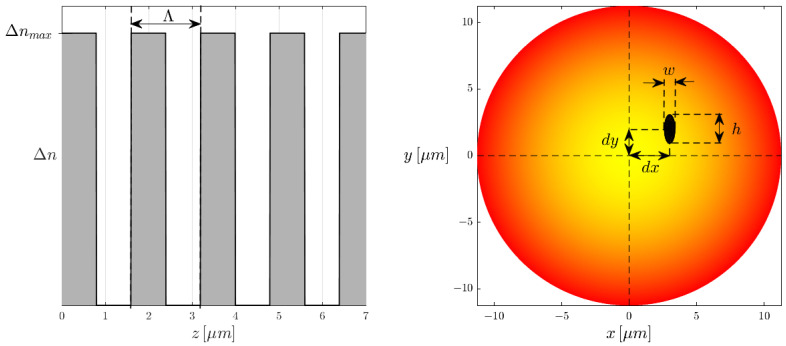
The longitudinal (**left**) and transverse (**right**) geometries of the localised point-by-point-inscribed FBG. The transverse RIM is represented by the black region, with this specific case depicting an eccentric FBG, i.e., dx=3μm and dy=2μm.

**Figure 12 sensors-25-06087-f012:**
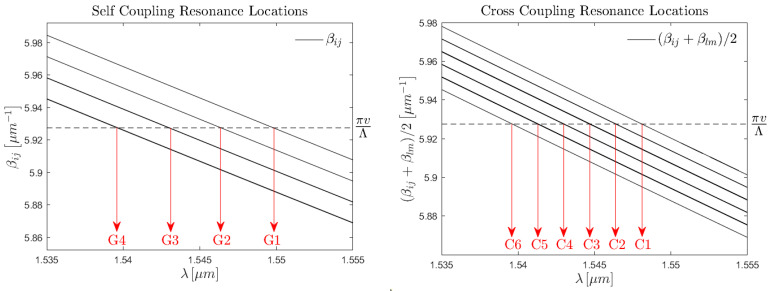
The self-coupling (**left**) and cross-coupling (**right**) resonance wavelength locations. Calculated as the wavelengths of maximum coupling according to the FBG phase matching condition.

**Figure 13 sensors-25-06087-f013:**
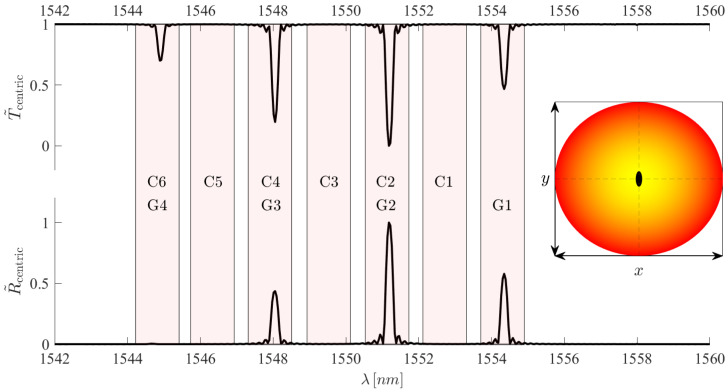
The simulated normalised FBG transmission and reflection spectra for a centric FBG inscribed at dy=0μm and dx=0μm. The transmission and reflection spectra have been normalised with regards to the minimum transmissivity and maximum reflectivity, respectively. The mode group self-coupling peaks are noted as G1 to G4, and the cross-coupling peaks noted as C1 to C6.

**Figure 14 sensors-25-06087-f014:**
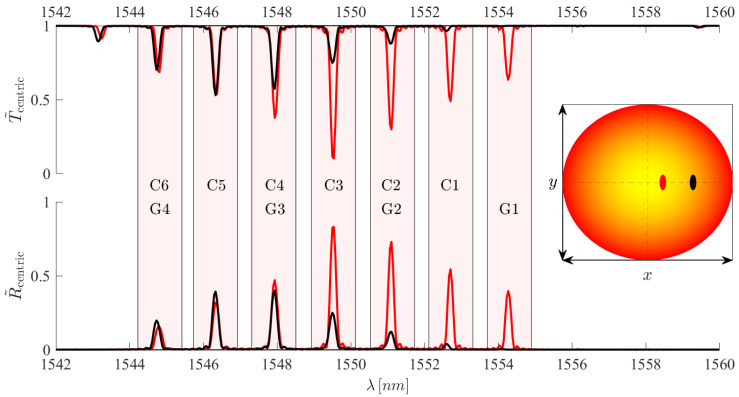
The simulated normalised FBG transmission and reflection spectra for two eccentric FBGs. The first inscribed at dy=0μm and dx=2μm (red line) and the second inscribed at dy=0μm and dx=6μm (black line). The transmission and reflection spectra have been normalised with regards to the minimum transmissivity and maximum reflectivity of the spectra in Figure 13, respectively. The mode group self-coupling peaks are noted as G1 to G4, and the cross-coupling peaks noted as C1 to C6.

**Figure 15 sensors-25-06087-f015:**
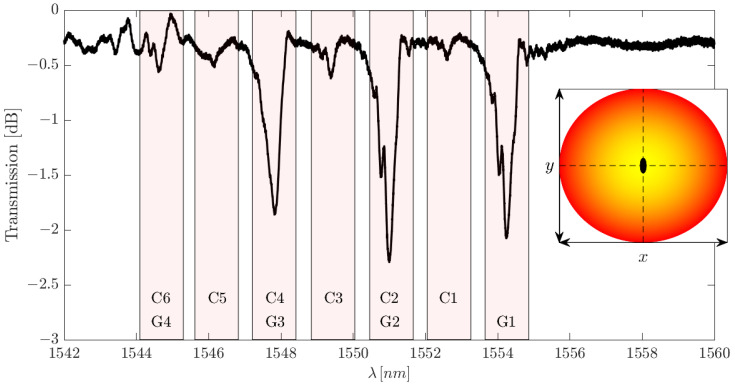
An experimentally obtained centric FBG inscribed at dy=0μm and dx=0μm. This FBG transmission spectrum was aquired in the absence of the spatial mode multiplexer and was connected directly to the circulator of Figure 1. The mode group self-coupling peaks are noted as G1 to G4, and the cross-coupling peaks noted as C1 to C6.

**Figure 16 sensors-25-06087-f016:**
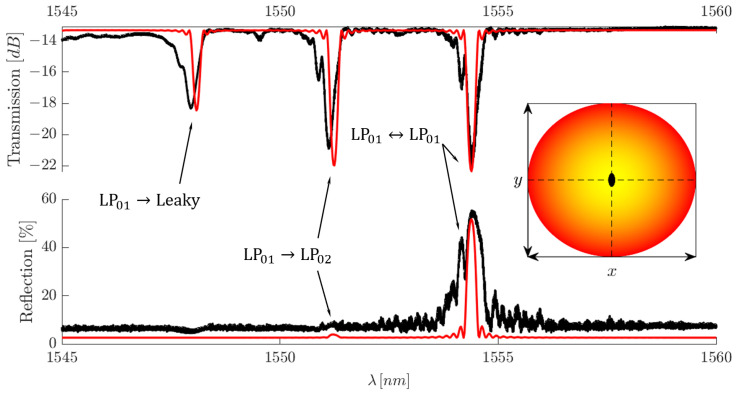
Experimental (black line) and simulation (red line) transmission (**top**) and reflection (**bottom**) spectra for a centric FBG, dy=dx=0μm, excited solely by HG00. The inset depicts the FBG transverse location within the fibre core.

**Figure 17 sensors-25-06087-f017:**
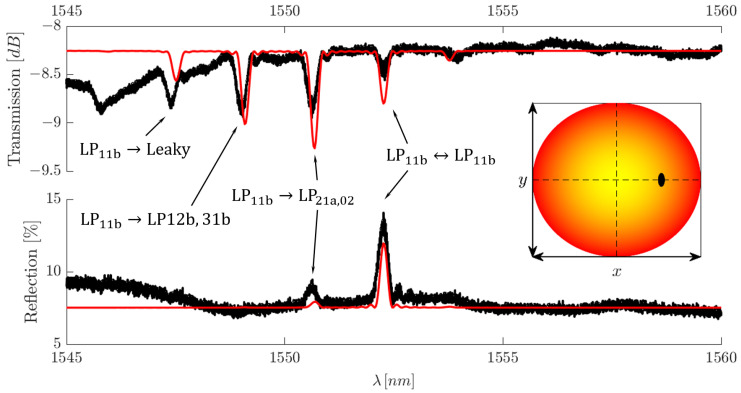
Experimental (black line) and simulation (red line) transmission (**top**) and reflection (**bottom**) spectra for an eccentric FBG, dy=0μm, dx=6μm, excited solely by ${\mathrm{HG}}_{01}$. The inlay depicts the FBG transverse location within the fibre core.

**Figure 18 sensors-25-06087-f018:**
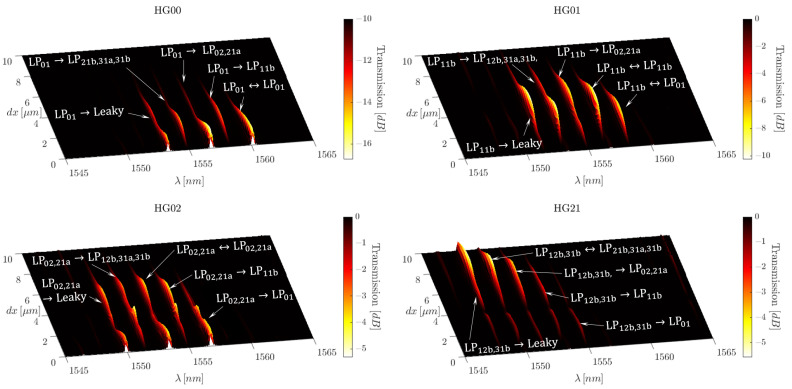
Simulated transmission density plots for four different HG mode initial excitations. The y-axis, dx represents different FBG eccentricities in the y-axis while the y-eccentricity is fixed at dy=0μm. Therefore, each layer along the y-axis gives the transmission spectrum for the given value of dx.

**Figure 19 sensors-25-06087-f019:**
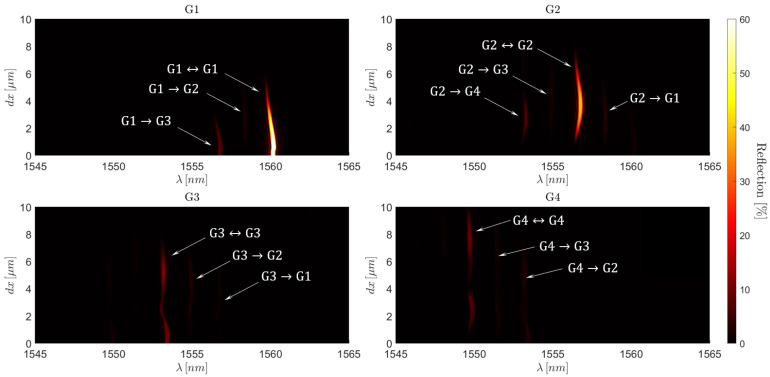
Simulated reflection density plots for all four mode groups. Each group consists of a summation of the individual reflection spectra for each mode excitation in each group. Here, a common heatmap has been applied to highlight the relative differences in reflectivity between mode groups for a fixed length of 5 mm and an arbitrary $\Delta n_{\max}$ for the FBG.

**Table 1 sensors-25-06087-t001:** Proteus output and Prysmian low DMDG 10 LP mode fibre-mode group composition ^*a*^.

Group Number	HG Output Modes	LP Fibre Modes
g=1	$HG_{00}$	$LP_{01}$
g=2	$HG_{01}$, $HG_{10}$	$LP_{11a}$, $LP_{11b}$
g=3	$HG_{11}$, $HG_{02}$, $HG_{20}$	$LP_{02}$, $LP_{21a}$, $LP_{21b}$
g=4	$HG_{12}$, $HG_{21}$, $HG_{03}$, $HG_{30}$	$LP_{12a}$, $LP_{12b}$, $LP_{31a}$, $LP_{31b}$

^*a*^ The leaky-mode group has been omitted here.

## Data Availability

The original contributions presented in this study are included in the article. Further inquiries can be directed to the corresponding authors.

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
