# Peer review of "Numerical and Experimental Study of Mode Coupling Due to Localised Few-Mode Fibre Bragg Gratings and a Spatial Mode Multiplexer"

_sensors, 2025, doi:10.3390/s25196087_

Round 1
Reviewer 1 Report
Comments and Suggestions for Authors
Dear authors of the paper "Numerical and Experimental Study of Mode Coupling due to Localised Few-mode Fibre Bragg Gratings and as Spatial Mode Multiplexer" please attend the following observations:
- In line 105, explain why a photonic lantern which is also a spatial mode multiplexer cannot be used in this application.
- In line 111, explain why a 5 m extension fibre is needed.
- In line 120, explain which method or instrument was used to measure the refractive index profile of the few-mode fibre.
- In line 170 it says that to maintain the localised nature of the FBG RIM, it was written via a single-passing of the laser and in line 172 it says that there is a formation of multiple modifications with each re-passage of the laser. Explain when single-passing and multiple re-passages are used.
- Around line 689, provide an example of how spatial mode multiplexing with two localised FBGs of the same period (one centric and the other highly eccentric) could be used in a real sensor.
Best Regards
Author Response
Firstly, thank you for your detailed response. We hape we have been able to address all of your comments in the breakdown below. Additionally, please see attached a pdf of the modified manuscript with all changes highlighted in orange. This includes changes from all reviewers.
Comment 1: In line 105, explain why a photonic lantern which is also a spatial mode multiplexer cannot be used in this application.
This comment, along with comments given by reviewer 2, sparked us to modify section 2.1. In this section we have added a breakdown of the criteria a spatial mode multiplexer must fit in order to work effectively within out proposed setup. This is in order to point out that MPLC based mode multiplexers are not the sole candidate for this application. Following this breakdown, we propose two key candidates to replace the Proteus, namely, a specific case of a photonic lantern and free space mode multiplexers based on phase plates. We invite you to read through this adapted section 2.1 with hopes that it addresses your comment.
Comment 2: In line 111, explain why a 5 m extension fibre is needed.
The following paragraph has been added for clarity…
“To protect the jacketed 10 m output fibre from damage during testing, a 5 m unjacketed extension fibre was spliced at its end face. Since many FBGs with varying eccentricities were to be tested, the output fibre would have to undergo frequent cutting, cleaving, and splicing. The extension fibre then served as an expendable length of fibre, in place to absorb this splice-induced degradation. Thus, preserving the integrity of the main output fibre, an integrated part of the Proteus device”.
Comment 3: In line 120, explain which method or instrument was used to measure the refractive index profile of the few-mode fibre.
The requested detail has been added to the text and now reads…
“Therefore, for accurate numerical modelling, the refractive index profile was obtained experimentally using an IFA-100 interferometric fibre analyser operating at 633 nm. This specific instrument allows for the refractive index profile to be determined with an accuracy of ±0.0001 with a spatial resolution of 500 nm. This refractive index profile, for the specific batch of the fibre in our possession, was then used in all following numerical studies…”.
Comment 3: In line 170 it says that to maintain the localised nature of the FBG RIM, it was written via a single-passing of the laser and in line 172 it says that there is a formation of multiple modifications with each re-passage of the laser. Explain when single-passing and multiple re-passages are used.
I believe there has been a misunderstanding as to what we were trying to explain in this section of the text. Therefore, this explanation of the inscription constraints has been adapted to make sure this section reads better and will avoid any future confusion. The new text reads as follows…
“To maintain the localised nature of the FBG refractive index modification (RIM), the laser pulse energy was limited to a maximum value of 7.94 μJ. This helped to avoid laser self-focussing effects evoked by high pulse energies, Fig. 3(c). Additionally, all FBGs in this study were inscribed via a single passage of the laser to prevent the formation of multiple RIMs. We observed that multiple RIMs often formed when using multiple laser passes, a phenomenon we believe to be caused by shifts in the focal point between consecutive re-passages, potentially due to the laser-induced densification of the silica, Fig. 3(d)…”.
Comment 3: Around line 689, provide an example of how spatial mode multiplexing with two localised FBGs of the same period (one centric and the other highly eccentric) could be used in a real sensor.
The following text has been added to address this comment…
“This new form of FBG multiplexing has a direct implication for FBG sensing arrays. If the concept is scaled up to arrays of FBGs, with their periods distinctly organised for WDM, one array can be inscribed centric, and the other highly eccentric. By doing so we obtain a way to effectively double the number of FBGs in the WDM array.
In its standard single-mode FBG configuration, WDM is limited by the interrogator bandwidth and the wavelength separation between adjacent Bragg peaks. This ultimately couples together the dynamic range of each FBG and the maximum number of FBGs that can be included in an array. Thus, if one were to increase, the other must be decreased. A common method to increase the multiplexing capability of WDM arrays is to employ the use of TDM. However, this method depends on complex synchronisation techniques to function correctly and is therefore strongly affected by any jitter or computer lag in the interrogation system.
Furthermore, the nature of the TDM method forces a minimum distance between FBGs of the same period to avoid signal overlap. This then impacts sensor spacing and may hinder the formation of dense quasi-distributed sensor arrays requiring a large number of sensors. Therefore, our presented method may provide an alternative to TDM for increasing the capacity of FBG sensing arrays distributed using the WDM method”.
Thank you again for your time.
Kind regards,
James Hainsworth

Reviewer 2 Report
Comments and Suggestions for Authors
The paper proposes and presents a new experimental approach for implementation of mode coupling to few -mode fibre Brag grating by employing a spatial mode multiplexer. The work considers the PbP Bragg grating inscription and studies conditions for selective fiber modes excitation. Notably, the results show that the introduction and the specific placement of a spatial mode multiplexer leads to slight simplification of the FBG transmission spectrum due to modal filtering at the reflected light, suggesting thus potential wavelength division multiplexing (WDM) monitoring of FM-FBGs. The study is integrated with a detailed theoretical and numerical analysis of the modal excitation and mode transformation supporting the experimental results and gaining valuable theoretical insight.
Overall, the paper presents high quality work, well validated by combining tightly connected experimental and theoretical approaches, providing also well described techniques of education also value to the generic audience.
The proposed approach could be of interest to FBG based sensing approaches and lies within the thematic priorities of the Journal.
Although valid, the theoretical explanation and connection to experimental findings is somehow complex, and some questions can be raised.
- The employed spatial mode multiplexer Cailabs Proteus-S can operate (according to its specs) at different modes such as : HG, LP OAM, LG. Using the LP excitation mode the experimental and theoretical analysis (despite its value) would be much simpler. Indeed, authors comment at 6.1 section: “It could be very interesting to investigate a FM-fibre with less supported modes with a spatial mode multiplexer that works directly in LP modes. This will allow for further refining of our numerical model under these simplified circumstances, and to see if a fibre that supports fewer modes will prove a better candidate for multiplexing via mode selectivity”. Why the LP excitation mode was not used then? On the other hand the HG allows the simultaneous excitation of multiple LP modes which is important for the multiple mode coupling demonstration. Could authors please comment and clarify?
- The employed Cailabs Proteus-S spatial mode multiplexer is a commercial equipment and extra care should be taken when directy refer to a commercial equipment with such a central role in a scientific paper. Maybe the Journal has some relevant guidelines or recommendations. Authors could provide a brief discussion on alternative spatial mode multiplexers, and provide a valid comparison stating the required specifications of such instruments in a systematic -scientific way.
- Although the approach is valid and interesting there are concerns regarding the complexity and reliability of the method. Most importantly on the requirements on the repeatability of writing such accurately positioned BG in the fiber. Furthermore, the inscription and operation of third order BGs limits their absolute strength thus limiting their reliability or operability in noisy applications.
Author Response
Firstly, thank you for your detailed response. We hape we have been able to address all of your comments in the breakdown below. Additionally, please see attached a pdf of the modified manuscript with all changes highlighted in orange. This includes changes from all reviewers.
Comment 1: The employed spatial mode multiplexer Cailabs Proteus-S can operate (according to its specs) at different modes such as : HG, LP OAM, LG. Using the LP excitation mode the experimental and theoretical analysis (despite its value) would be much simpler. Why the LP excitation mode was not used then? On the other hand the HG allows the simultaneous excitation of multiple LP modes which is important for the multiple mode coupling demonstration. Could authors please comment and clarify?
Comment 2: The employed Cailabs Proteus-S spatial mode multiplexer is a commercial equipment and extra care should be taken when directy refer to a commercial equipment with such a central role in a scientific paper. Maybe the Journal has some relevant guidelines or recommendations. Authors could provide a brief discussion on alternative spatial mode multiplexers, and provide a valid comparison stating the required specifications of such instruments in a systematic -scientific way.
We respond directly to comments 1 and 2, as well as Reporter 1’s comment 1, by adapting the text in section 2.1 and re-organising for readability. We invite you to read the corrected section. To summarise, we explain that the Proteus designed for HG excitation was the device available to us at the time of the investigation. Furthermore, a paragraph has been added discussing the key criteria a spatial mode MUX should have to be functional in the proposed setup, as well as providing two strong candidates for use.
Comment 3: Although the approach is valid and interesting there are concerns regarding the complexity and reliability of the method. Most importantly on the requirements on the repeatability of writing such accurately positioned BG in the fiber. Furthermore, the inscription and operation of third order BGs limits their absolute strength thus limiting their reliability or operability in noisy applications.
These concerns are understood and somewhat valid for low accuracy inscription setups. Nevertheless, highly accurate and rigorously calibrated commercial inscription setups are designed to inscribe the FBG as close to the transverse centre of the fibre core as possible. This is done in order to limit losses caused by cladding mode coupling in SMFs. Highly eccentric FBGs can be easily created by applying a dx or dy offset in the inscription software. While this seems a challenging inscription configuration, nanometric x,y,z stages means repeatability is obtainable.
To address this final point about third order FBGs limiting their absolute strength, this is indeed a very valid point. However, second order FBGs are obtainable when using objectives with a higher numerical aperture. Furthermore, the FBGs inscribed in this work are all limited to a constant length of 5mm. Increasing the FBG length would also help increase reflectivity, improving usability for noisy applications. Following comments from reviewer 3, we have included a new section, numbered 5.5. Here we open the discourse around FBG geometry and reflectivity, both in the longitudinal and transverse directions.
Thank you again for your time,
Kind regards,
James Hainsworth

Reviewer 3 Report
Comments and Suggestions for Authors
The manuscript presents an exhaustive and complete study and analysis of mode conversion effects when FBGs are inscribed in Few-Mode (FM) optical fibers. The work considers point-to-point inscribed FBGs differentiating three locations with respect eccentricity, i. e. centric, low and high eccentricity. Furthermore, the work introduces a spatial mode multiplexer with the objective to control the mode excitation and hence, he mode coupling. The manuscript is well structured and well written.
However, there are some points that should be discussed in order to clarify in advantage this manuscript. Please find enclosed below the list of detailed remarks.
Detailed Remarks
- A better explanation for experimental setup presented in Fig. 1 is necessary in order to identify how the FBG is excited (with single HG mode input selection) and interrogated, and how the reflection and transmission spectra are measured. (Is it correct the sense of the arrow inside the circulator?).
- It would be desirable to add the geometric design and refractive index profile for the Prysmian 6LP mode fibre, before numerical study theory for LP modes.
- It would desirable to explain where the HG modes are generated with the purpose to excite the core guided LP modes.
- The developed study is applied to FBGs type II ellipse with a heigh of 2.2um and a width of 0.8um. What happened with other FBG type II dimensions? A discussion must be taken into consideration.
- 13 should represent the comparison between experimental and simulated results for the eccentric FBG, both for reflection and transmission spectra.
- In Fig. 14 and 15, it is neccessary to identify the experimental data (black line?) for the simulation data (red line?)
- A deep explanation for the proposal to use the selective modal excitation, selective filtering, and FBG eccentricity to multiplex FBGs would be necessary.
- A final example to use this FBG multiplexing technique in applications such as sensing or fiber laser would be advisable.
Author Response
Firstly, thank you for your detailed response. We hope we have been able to address all of your comments in the breakdown below. Additionally, please see attached a pdf of the modified manuscript with all changes highlighted in orange. This includes changes from all reviewers.
Comment 1: A better explanation for experimental setup presented in Fig. 1 is necessary in order to identify how the FBG is excited (with single HG mode input selection) and interrogated, and how the reflection and transmission spectra are measured. (Is it correct the sense of the arrow inside the circulator?).
We have addressed this comment by adapting the section 2.1. Following additional comments from reviewer 1 and 2 this section has been completely reworked.
Secondly, thank you for pointing out the inaccuracy regarding the circulator arrow direction. This was indeed the wrong sense. We have corrected this error and reverted its direction.
Comment 2: It would be desirable to add the geometric design and refractive index profile for the Prysmian 6LP mode fibre, before numerical study theory for LP modes.
We have taken this into account and moved this section. The new section number is, section 3.1.
Comment 3: It would desirable to explain where the HG modes are generated with the purpose to excite the core guided LP modes.
Within the re-written section 2.1 we explain that the HG modes are generated within the Proteus. These free space modes are then coupled to the fibre by projecting them on the fibre LP modes, again within the device itself. We also explain that the Proteus can be made to specification to generate other modes, however the Proteus available to us at the time of the investigation was designed for HG operation.
Furthermore, later in the manuscript we explain how the HG modes are coupled to the fibre as a superposition of LP modes.
Comment 4: The developed study is applied to FBGs type II ellipse with a heigh of 2.2um and a width of 0.8um. What happened with other FBG type II dimensions? A discussion must be taken into consideration.
This comment is entirely relevant and we believe a very interesting study to conduct. However, detailed analysis on how the FBG geometry effects the spectral output would be a very dense study and would merit its own article. Nevertheless, we agree that opening a discourse in a new section, just before the conclusion discussing this topic would certainly be interesting. Please see this new section 5.5 where we begin a short discussion about FBG geometry. We also discuss the FBG length and order following comments from reviewer 2.
Comment 5: 13 should represent the comparison between experimental and simulated results for the eccentric FBG, both for reflection and transmission spectra.
Thank you for this comment, we certainly understand these thoughts. However, there are several problems with this idea and we propose an alternative below. Figure 13 represents a low eccentricity FBG that hasn’t been selectively excited. Meaning the spectrum was obtained by connecting the FBG fibre directly to the circulator in figure 1. The proteus has been bypassed and the transmission spectrum taken. Two problems arise here, firstly, the reflected signal has to pass from few-mode to single mode. This results in multi-mode interference, making the reflection spectrum chaotic and it become difficult to highlight any spectral features. Sadly, no commercial circulators have been made in the Prysmian fibre used in this study. Therefore, the reflection spectrum can’t be compared. Secondly, as the FMF hasn’t undergone selective excitation, we have no way of knowing how the energy is distributed between modes. Therefore, to obtain a similar spectral output we would have to randomly change the modal power distribution in our simulation, thus forcing the spectra to match. To us, this doesn’t constitute a real validation.
However, we propose the introduction of two simulated figures. These figures will show the mode groups as before, to give a visual aid on the spectral features of the FBG. The new figure will present the three cases, centric, low eccentricity and high eccentricity, and will include both the reflection and transmission spectra. These spectra will represent the purely theoretical case when then mode have all been excited evenly, possessing each 10% of the energy coupled to the fibre. The model validation then comes in the next section where we can directly compare the results as we have a better understanding of the modal power distribution in the fibre. However, we provide an additional figure, with a centric FBG which is excited directly by the circulator, bypassing the Proteus. This is added as a direct comparison to the simulated data, in order to strike an initial comparison before the validation is carried out. We hope this satisfies your comments.
Comment 6: In Fig. 14 and 15, it is neccessary to identify the experimental data (black line?) for the simulation data (red line?)
Thank you for noticing this error. The experimental and simulated lines have now been identified in the figure captions. They now read as follows…
“Experimental (black line) and simulation (red line) transmission (top) and reflection (bottom) spectra for…”
Comment 7: A deep explanation for the proposal to use the selective modal excitation, selective filtering, and FBG eccentricity to multiplex FBGs would be necessary.
Comment 8: A final example to use this FBG multiplexing technique in applications such as sensing or fiber laser would be advisable.
I believe we have responded to these two comments simultaneously by adding an explanation of how this can help increase the capacity of WDM sensing arrays. This was initially added in response to reviewer 1’s comment 5.
The following text has been added to address this comment…
“This new form of FBG multiplexing has a direct implication for FBG sensing arrays. If the concept is scaled up to arrays of FBGs, with their periods distinctly organised for WDM, one array can be inscribed centric, and the other highly eccentric. By doing so we obtain a way to effectively double the number of FBGs in the WDM array.
In its standard single-mode FBG configuration, WDM is limited by the interrogator bandwidth and the wavelength separation between adjacent Bragg peaks. This ultimately couples together the dynamic range of each FBG and the maximum number of FBGs that can be included in an array. Thus, if one were to increase, the other must be decreased. A common method to increase the multiplexing capability of WDM arrays is to employ the use of TDM. However, this method depends on complex synchronisation techniques to function correctly and is therefore strongly affected by any jitter or computer lag in the interrogation system.
Furthermore, the nature of the TDM method forces a minimum distance between FBGs of the same period to avoid signal overlap. This then impacts sensor spacing and may hinder the formation of dense quasi-distributed sensor arrays requiring a large number of sensors. Therefore, our presented method may provide an alternative to TDM for increasing the capacity of FBG sensing arrays distributed using the WDM method”.
Thank you again for your time.
Kind regards,
James Hainsworth
